# RankGen: A Statistically Robust Framework for Ranking Generative Models Using Classifier-Based Metrics

## Abstract

Standard metrics for evaluating generative models are brittle, easy to game, and often ignore task relevance. We introduce RankGen, a unified evaluation framework built on four metrics: Quality, Utility, Indistinguishability, and Similarity; each designed to capture a distinct failure mode and supported by PAC-style generalization bounds. RankGen follows a two-stage process: models that violate bounds are discarded, while the rest are ranked using robust, quantile-based summaries. The resulting composite score, Exchangeability, captures both fidelity and task relevance. By exposing hidden pathologies such as memorization, RankGen provides a principled foundation for safer model selection and deployment.

## 1 Introduction

Generative models are rapidly becoming core components of modern machine learning pipelines. They power text assistants, accelerate drug discovery, and enable creative applications across images, audio, and code. Their reach means the reliability of generative systems is no longer an academic concern: it directly shapes the safety, fairness, and utility of downstream applications. Yet despite the dramatic progress in generative capability, one question remains unresolved: *how should we evaluate generative models in a way that is both rigorous and actionable?*

Flawed evaluation protocols can reward models that memorize training data, masking privacy violations behind inflated similarity scores. They can overlook subtle distributional shifts that later collapse downstream performance, or promote models that look impressive to human inspection while offering no measurable utility in task-centric pipelines. Conversely, a sound evaluation framework would expose such failure modes and guide principled model selection in settings where human curation is infeasible.

Meeting this bar demands treating generator quality as multi-faceted. Fidelity must be distinguished from diversity; surface resemblance from task relevance; broad distribution alignment from the generated samples fit among their real nearest neighbours. Failures along any of these axes can derail deployment even when headline scores look strong. Figure 1 previews these pathologies, framing evaluation as a search for complementary probes rather than a single scalar.

Existing heuristics rarely satisfy these requirements. Popular scalar scores and ad-hoc summaries collapse diverse behaviours into one number, providing little diagnostic value and remaining brittle to seemingly innocuous design choices. While Sec. 2 surveys alternatives, most lack principled uncertainty estimates and are easy to game.

We therefore advocate reframing evaluation as *diagnosis.*

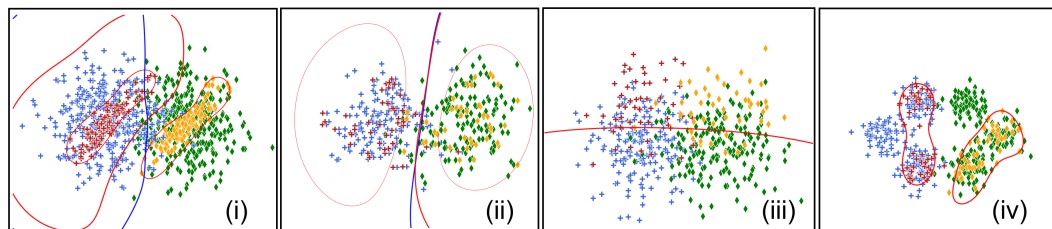

Figure 1: Four common generative failure modes—and how RankGen detects them. Each panel shows real data (blue and green) and generated data (red and orange) for a binary classification task. Marker shapes denote classes. (i) Quality – Poor task fidelity: The generated data lies in dense regions of the real distribution but fails to produce a decision boundary that generalizes. A classifier trained only on generated data underperforms, revealing a fidelity gap. (ii) Utility – No added value: Generated points are near-copies of real data. When added to real training data, they do not shift the decision boundary or improve performance. Useful generative data should meaningfully expand the training set. (iii) Indistinguishability – Easy to tell apart: A subtle tell-tale (e.g., a shift in a single feature) allows a classifier to distinguish generated from real. While the two distributions may appear similar overall according to the majority of features, a discriminator easily separates them. (iv) Similarity – Local mismatch: In a well-behaved model, the neighbourhood around any instance should contain a balanced mix of real and generated samples. When neighbourhoods contain only one type—either real or generated—it signals poor mixing, reflecting structural differences or mode collapse. Neighbourhood entropy quantifies this mismatch, even when the global distributions appear to overlap.

**Our approach and contributions.** We present **RankGen**, a general framework for evaluating generative models that goes beyond fragile one-number scores. RankGen looks at models from four different angles: *Quality* (does the model capture the true signal?), *Utility* (does it add useful new data?), *Indistinguishability* (can real and generated data be told apart?), and *Similarity* (do real and generated data mix well at a local level?). Instead of relying on raw numbers alone, RankGen attaches statistical guarantees to each measure, so results are trustworthy rather than misleading.

The evaluation runs in two stages. First, models that clearly fail the PAC statistical checks are filtered out. Second, the remaining models are compared using robust summaries that take uncertainty into account. A final combined score, called *Exchangeability*, balances predictive value and distributional alignment, making sure that strength in one area cannot hide weakness in another.

We test RankGen on both controlled experiments (e.g., mode collapse, label noise, copying) and real datasets (molecules and images). The framework consistently uncovers problems—especially memorization—that standard metrics overlook. More than just ranking models, RankGen provides an interpretable profile of *how* each model falls short, helping guide safer choices and deployments in settings where human inspection is not feasible.

## 2 Background and Related Work

**Scalar heuristics.** Evaluation has long been dominated by scalar scores such as IS (Salimans et al., 2016), FID (Heusel et al., 2017), and kernel distances like MMD/KID (Gretton et al., 2012; Binkowski et al., 2018). These metrics offer convenient one-number comparisons but rest on restrictive assumptions, are sensitive to feature choice and preprocessing (Barratt and Sharma, 2018; Chong and Forsyth, 2020; Parmar et al., 2022; Betzalel et al., 2022), and often fail to track human judgment or downstream utility (Theis et al., 2016; Borji, 2019). Empirical studies show that small changes in embedding or sample size can flip rankings, and the resulting scores provide little diagnostic insight. **Multi-dimensional diagnostics.** To capture more structure, precision–recall curves for distributions (Sajjadi et al., 2018; Kynkäänniemi et al., 2019), density–coverage (Naeem et al., 2020), and more recent measures such as VENDI and FKEA (Friedman and Dieng, 2023; Jalali et al., 2024)

separate fidelity from diversity. These methods detect issues like mode collapse that scalars miss, but they inherit fragility from their embeddings and lack distribution-free guarantees, limiting trustworthiness across domains (Parmar et al., 2022; Betzalel et al., 2022). **Classifier-based evaluation.** Protocols like GAN-train/test (Shmelkov et al., 2018) and TSTR (Hyland et al., 2017; Yoon et al., 2019) assess whether synthetic data supports downstream prediction. This connects evaluation to practical utility, but results are usually reported as point estimates without uncertainty, making it hard to know whether differences are significant. Moreover, they conflate distinct failure modes: poor fidelity, diversity loss, and copying can all produce similar outcomes. **Memorization and leakage.** Generators can achieve deceptively strong scores by memorizing training data. This inflates similarity-style metrics while adding no useful variation and raises privacy risks. Attacks on GANs and diffusion models have demonstrated practical data leakage (Hayes et al., 2019; Chen et al., 2020; Carlini et al., 2023), yet most evaluation protocols are not designed to expose it (Bhattacharjee et al., 2023). **Domain-specific benchmarks.** In molecules and graphs, benchmarks such as GUACAMOL and MOSES (Brown et al., 2019; Polykovskiy et al., 2020) emphasize validity, uniqueness, and novelty. These checks are useful but narrow: a model can satisfy them while merely reproducing known molecules. Broader graph generation often relies on plausibility heuristics or manual inspection (You et al., 2018), highlighting the lack of systematic, statistically grounded tools. Across domains, the field has moved from brittle scalars to fragile multi-dimensional measures and under-specified classifier probes. Memorization remains a persistent blind spot, and domain-specific metrics only partially address the problem. This fragmented landscape motivates the need for evaluation that is interpretable, statistically guaranteed, resistant to copying, and applicable across modalities.

# 3 THE METHOD: RANKGEN

**RankGen** evaluates generators not by surface similarity alone, but by combining downstream predictive utility and distributional alignment, both global and local. We unify evaluation into four classifier-based metrics—Quality, Utility, Indistinguishability, and Similarity—each equipped with a PAC-style generalization bound that links finite-sample estimates to their population counterparts with high confidence (see Appendix 7 for assumptions and constants). To balance *reliability* and *resolution*, bounds act as *gates* that filter out statistically invalid models, while robust quantile summaries (medians/IQRs) rank those that pass, capturing both the typical performance (via the median) and its variability across resamples (via the interquartile range).

Our procedure has four stages. (i) *Data preparation:* multiple stratified train/test splits with class-conditional generation under strict isolation. (ii) *Metric evaluation:* compute the four metrics per split with PAC bounds and quantile summaries. (iii) *Ranking:* filter models by bound violations, then order survivors using quantile summaries and pairwise dominance. (iv) *Diagnosis:* metric profiles reveal characteristic failure modes—e.g., high Similarity but low Utility (copying) or strong Quality but low Indistinguishability (global shifts)—providing interpretable fingerprints rather than opaque scores.

## 3.1 DATA PREPARATION AND SAMPLING

Consider a classification dataset $D = \{(x_i, y_i)\}_{i=1}^{N}$, where each feature vector $x_i \in \mathbb{R}^d$ lies in a $d$-dimensional space, and each label $y_i \in \{0, 1, \ldots, C\}$ denotes one of $C$ classes. We first partition $D$ into a training set $D_{\text{train}}$ and a held-out test set $D_{\text{test}}$. The training set is then further divided, using stratified sampling, into two equally sized subsets, $D_{\text{train1}}$ and $D_{\text{train2}}$.

**Training the generators.** We adopt a class-wise strategy for generating synthetic data. Unless the underlying model is inherently class-conditional, we train a separate generator for each class. Specifically, for every class $c \in C$, we fit a generative model $f_c$ using only the corresponding subset $D_{\text{train1}}^c$ of the training data. For models that are unconditional, we use the same generator architecture, but train it independently on the samples of each class.

**Constructing the synthetic pool.** After training, we draw synthetic samples once for each class to construct a base synthetic dataset. Let $p_{f_c}$ denote the distribution induced by the generator $f_c$. For each class $c$, we sample $\{x_{c,\ell}\}_{\ell=1}^{N_c} \sim p_{f_c}$, where $N_c = |D_{\text{train1}}^c|$ is the desired number of generated examples for class $c$. The resulting synthetic dataset is then the union of all class specific samples

$$D_{\text{generated}} = \bigcup_c \{(x_{c,\ell},\, c) : \ell = 1, \ldots, N_c\}.$$

**Uncertainty estimate.** To estimate uncertainty for each replicate $i$, we draw bootstrap samples $D_{\text{train1}}^{(i)} \subseteq D_{\text{train1}}, \quad D_{\text{train2}}^{(i)} \subseteq D_{\text{train2}}, \quad D_{\text{generated}}^{(i)} \subseteq D_{\text{generated}},$ using class-stratified bootstrapping. Each replicate is used to compute all four metrics and their quantile summaries.

## 3.2 Classifier Performance Scores ($\rho_i$)

We propose 4 key metrics: Quality, Utility, Indistinguishability, and Similarity defined via classifier-derived scores $\rho_j^{(i)}$ (Table 1), with every score corresponding to a distinct training-and-testing configuration on split $i$. All models are evaluated on the same shared test set $D_{\text{test}}$. Here, $\Phi_\theta$ denotes a task classifier, $\Psi_\theta$ a binary real-vs-generated discriminator, and $\Gamma_\theta$ a structure-aware similarity model (e.g., a data-specific kernel).

Table 1: Classifier-based evaluation metrics used in RankGen.

| Metric | Description |
|---|---|
| $\rho_1^{(i)} = \Phi_\theta(D_{\text{train1}}^{(i)})$ | Accuracy of classifier trained on real data only (baseline). |
| $\rho_2^{(i)} = \Phi_\theta(D_{\text{generated}}^{(i)})$ | Accuracy when trained on generated data only (signal fidelity). |
| $\rho_3^{(i)} = \Phi_\theta(D_{\text{train1}}^{(i)} \cup D_{\text{train2}}^{(i)})$ | Accuracy with extra real data (upper bound). |
| $\rho_4^{(i)} = \Phi_\theta(D_{\text{train1}}^{(i)} \cup D_{\text{generated}}^{(i)})$ | Accuracy with real+generated data (usefulness). |
| $\rho_5^{(i)} = \Psi_\theta(D_{\text{train1}}^{(i)}, D_{\text{generated}}^{(i)})$ | Discriminator accuracy distinguishing real vs. generated. |
| $\rho_6^{(i)} = \Gamma_\theta(D_{\text{train1}}^{(i)}, D_{\text{generated}}^{(i)})$ | Entropy-based similarity of local feature neighborhoods. |

## 3.3 Quality Metric

**Definition.** The quality score measures how much predictive signal synthetic data preserves relative to real data. For split $i$, let $\rho_1^{(i)}$ be the task classification score (e.g. accuracy, AUC, F1) of a classifier trained on $D_{\text{train1}}^{(i)}$ and tested on $D_{\text{test}}$, and $\rho_2^{(i)}$ the same score when trained on $D_{\text{generated}}^{(i)}$. We define

$$Q^{(i)} := \min\left(\frac{\rho_2^{(i)}}{\rho_1^{(i)}},\, 1\right) \in [0, 1].$$

**PAC bound.** Under PAC-learning assumptions, with probability $1 - \delta$,

$$Q^{(i)} \geq 1 - \frac{\varepsilon_1}{\rho_1^{(i)}},$$

where $\varepsilon_1 = 4\mathcal{R}_n + 2\sqrt{\frac{\ln(2/\delta)}{2n}}$ and $\mathcal{R}_n$ is the Rademacher complexity of the classifier class (derivation in App. 6).**Diagnostic role.** High $Q$ implies synthetic data supports task learning; low $Q$ indicates fidelity gaps such as noise or mode collapse.

## 3.4 Utility Metric

**Definition.** The utility score measures how much useful, nonredundant information synthetic data provides when augmenting real data. For split $i$, let $\Delta_{\text{real}}^{(i)} = \rho_3^{(i)} - \rho_1^{(i)}$ be the

gain from extra real samples and $\Delta_{\text{gen}}^{(i)} = \rho_4^{(i)} - \rho_1^{(i)}$ the gain from generated samples. We define

$$U^{(i)} := \min\left\{1, \; \max\left\{0, \; \frac{\Delta_{\text{gen}}^{(i)}}{\Delta_{\text{real}}^{(i)}}\right\}\right\}.$$

**PAC bound.** With probability $1 - \delta$,

$$U^{(i)} \geq 1 - \frac{\varepsilon_2}{\Delta_{\text{real}}^{(i)}},$$

where $\Delta_{\text{real}}^{(i)} = \rho_3^{(i)} - \rho_1^{(i)}$ and $\varepsilon_2$ is the composite error term defined in Eq. (A.10), which collects the empirical discrepancy, its PAC deviation, the ERM generalisation terms, and the joint optimal risk.

**Diagnostic role.** Low $U$ indicates memorization or redundancy; values near 1 indicate added task-relevant diversity.

3.5 INDISTINGUISHABILITY METRIC

**Definition.** For split $i$, let $\rho_5^{(i)}$ be the accuracy of a balanced real-vs-generated discriminator. The empirical indistinguishability score is

$$I^{(i)} := 1 - 2\left|\rho_5^{(i)} - 0.5\right| \in [0, 1],$$

where values near 1 indicate chance-level discrimination (good alignment).

Let $I^\star := 1 - 2|R^\star - 0.5|$ denote the population analogue, where $R^\star$ is the true discrimination accuracy of the optimal real-vs-generated classifier under the underlying data distributions. This quantity plays the role of the "population target" in the PAC bound.

**PAC bound.** For a discriminator class of VC dimension $d$ and an evaluation set of size $m$, with probability $1 - \delta$,

$$\left|I^{(i)} - I^\star\right| \leq 2\sqrt{\frac{2d\ln(2m) + \ln(8/\delta)}{m}}.$$

**Diagnostic role.** Low $I^{(i)}$ indicates artefacts or distribution shifts; high $I^{(i)}$ indicates good alignment.

3.6 SIMILARITY METRIC

**Definition.** For split $i$, we form a mixed dataset

$$D_{\text{mix}}^{(i)} := D_{\text{train1}}^{(i)} \cup D_{\text{generated}}^{(i)},$$

and attach a binary *domain label* to each point indicating whether it comes from the real set $D_{\text{train1}}^{(i)}$ or the generated set $D_{\text{generated}}^{(i)}$. For any $x \in D_{\text{mix}}^{(i)}$, let $\hat{p}_x$ be the fraction of neighbours from the same domain among its $k$-nearest neighbours in $D_{\text{mix}}^{(i)}$, and let

$$H(p) := -p\log p - (1-p)\log(1-p)$$

be the binary entropy.

We then define the empirical similarity score

$$S^{(i)} := \frac{1}{n} \sum_{x \in D_{\text{mix}}^{(i)}} \frac{H(\hat{p}_x)}{\log 2} \in [0, 1],$$

where $n = |D_{\text{mix}}^{(i)}|$ is the size of the mixed real+generated dataset.

**PAC bound.** Let $S^{(i)}$ denote the empirical similarity score above, and let $S^\star :=$ $\mathbb{E}_{x \sim D_{\mathrm{mix}}}[H(p_x)/\log 2]$ be its population analogue, where $p_x$ is the true in-domain neighbour proportion. With probability $1 - \delta$,

$$|S^{(i)} - S^\star| \leq \frac{1}{\log 2} \left( L_H \sqrt{\frac{\log(2n/\delta)}{2k}} + \log 2 \sqrt{\frac{\log(2/\delta)}{2n}} \right).$$

**Diagnostic role.** High $S^{(i)}$ indicates interleaving neighbourhoods of real and generated samples; low $S^{(i)}$ reveals mode collapse or geometric shift.

### 3.7 Exchangeability Metric

**Definition.** For replicate $i$, we combine predictive value (quality $Q^{(i)}$, utility $U^{(i)}$) and distributional alignment (indistinguishability $I^{(i)}$, similarity $S^{(i)}$). We form block averages

$$\Delta_{\mathrm{qual}}^{(i)} = \tfrac{1}{2}\Big(Q^{(i)} + U^{(i)}\Big), \qquad \Delta_{\mathrm{sim}}^{(i)} = \tfrac{1}{2}\Big(I^{(i)} + S^{(i)}\Big),$$

and define the conservative composite

$$\mathcal{E}_{\mathrm{min}}^{(i)} = \min\Big\{\Delta_{\mathrm{qual}}^{(i)}, \Delta_{\mathrm{sim}}^{(i)}\Big\} \in [0, 1].$$

**PAC guarantee.** If each base metric satisfies $M^{(i)} \geq 1 - \varepsilon_M$ with probability $1 - \delta_M$, then

$$\mathcal{E}_{\mathrm{min}}^{(i)} \geq 1 - \max\big\{\tfrac{1}{2}(\varepsilon_Q + \varepsilon_U), \ \tfrac{1}{2}(\varepsilon_I + \varepsilon_S)\big\},$$

under the allocation $\delta_Q = \delta_U = \delta_I = \delta_S = \delta/4$ and a union bound. This yields a composite guarantee holding with confidence at least $1 - \delta$ (App. 11).**Diagnostic role.** $\mathcal{E}_{\mathrm{min}}^{(i)}$ acts as a fail-safe summary: copying yields high $S^{(i)}$ but low $U^{(i)}$; models that improve accuracy but are easily detected score low on alignment. We report $\mathcal{E}_{\mathrm{min}}^{(i)}$ as the headline number and use the tuple $(Q^{(i)}, U^{(i)}, I^{(i)}, S^{(i)})$ to diagnose weaknesses.

### 3.8 Quantile-Based Estimation

Performance of each metric across resampled splits is frequently heavy–tailed or multimodal, so simple mean±std summaries are unstable. We therefore describe every generator–metric pair using empirical quartiles: the first quartile $Q_1$, median $M$, and third quartile $Q_3$. The median together with the interquartile range (IQR $= Q_3 - Q_1$) serves as a robust summary of central tendency and dispersion, remaining well behaved under outliers and skew.

For methods (including our Monte Carlo ranking in Sec. 3.9) that expect a mean and variance, we do not use the raw sample mean/std. Instead, we form a *pseudo*-mean and *pseudo*-standard deviation from the quartiles using the quantile-to-moment rules of Wan et al. (2016):

$$\hat{\mu}_Q = \tfrac{Q_1 + M + Q_3}{3}, \qquad \hat{\sigma}_Q = \frac{\mathrm{IQR}}{1.34898},$$

and use $(\hat{\mu}_Q, \hat{\sigma}_Q)$ solely as robust inputs to the ranking procedure and moment-based baselines, while reporting median/IQR as our primary summaries.

### 3.9 Ranking Across Metrics

RankGen adopts a two-stage ranking procedure that balances statistical reliability with resolution. In the first stage, we apply PAC-style bounds to each generator-metric pair to determine whether the empirical score exceeds a theoretical threshold with high confidence. This yields a binary pass/fail outcome for each metric (as shown in Table 2), allowing us to group generators by the number of metrics they satisfy. Generators that fail fewer bounds are considered more trustworthy and are placed in higher-priority groups.This hybrid strategy ensures that only statistically validated models are ranked, while still distinguishing subtle differences among high-performing generators. Once this filtering is complete, we

perform fine-grained ranking within each group using the quantile summaries introduced in Sec. 3.8. For each generator, the median acts as the location parameter and the IQR is converted into a pseudo standard deviation. We sample from a Gaussian with these parameters, truncate draws to the valid $[0, 1]$ range, and add a small ridge to the variance so that generators with nearly identical quartiles still participate. Repeating this Monte Carlo step produces synthetic metric profiles from which we compute pairwise dominance scores—the frequency with which one generator outperforms another across all metrics. Aggregating these dominance counts yields the final, uncertainty-aware ranking among statistically equivalent models. The algorithmic details are provided in App. 5.

| Generator | Quality | Utility | Indistinguish ability | Similarity | Exchange- ability |
|---|---|---|---|---|---|
| s4dd_u_1 | ✓ | ✓ | ✓ | ✓ | ✓ |
| s4dd_u_m | ✓ | ✓ | ✓ | ✓ | ✓ |
| stgg | ✓ | ✓ | ✓ | ✓ | ✓ |
| ns1 | ✓ | ✗ | ✓ | ✓ | ✗ |
| s4dd_f_m | ✗ | ✗ | ✓ | ✓ | ✗ |
| ns2 | ✓ | ✗ | ✓ | ✗ | ✗ |
| ns3 | ✓ | ✗ | ✓ | ✗ | ✗ |
| wgan | ✓ | ✗ | ✓ | ✗ | ✗ |
| gdss | ✗ | ✗ | ✓ | ✗ | ✗ |
| hiervae | ✗ | ✗ | ✓ | ✗ | ✗ |
| jtnn | ✗ | ✗ | ✓ | ✗ | ✗ |
| moflow | ✗ | ✗ | ✓ | ✗ | ✗ |
| swingnn | ✗ | ✗ | ✓ | ✗ | ✗ |

Table 2: Pass/fail checklist for each generator with respect to the PAC-style bounds of the four base metrics and the composite *Exchangeability*. A blue ✓ indicates the empirical score exceeds its theoretical lower bound (pass); a red ✗ indicates a violation. Full details of the datasets and generator variants appear in Sec. 4.2.

## 4    Empirical Evaluation

**Experimental scope and protocol.**   Our empirical study spans three settings: (i) *Synthetic* data, where we inject controlled perturbations (label noise, class imbalance, mode collapse, exact/noisy copies, Gaussian corruption) to stress–test metric sensitivity (Sections 4.1); (ii) *Molecular graphs* from the *Therapeutics Data Commons* (Huang et al., 2021) across AMES, BBB, CYP1A2, CYP2C19, hERG, and Lipophilicity (Section 4.2); and (iii) *Images* on MNIST (LeCun et al., 1998), Fashion–MNIST (Xiao et al., 2017), and CIFAR–10 (Krizhevsky, 2009) (Section 4.3). Across all settings we use class–conditional generation with strict train/test isolation, repeated stratified resampling, the same task classifier architecture, and median/IQR reporting. Implementation details, hyperparameters, and additional analyses are available in Appendix 14 and Appendix 22. Every metric is computed on held-out evaluation sets that are never touched during generator training or vectorizer fitting; auxiliary quantities such as $\widehat{\text{disc}}$ or $\widehat{\lambda}$ use disjoint resamples (cross-fitting) so that independence assumptions in the PAC derivations remain valid.

**Diagnostic framing.**   Our goal is not to produce a leaderboard, but to demonstrate RankGen's diagnostic power. Synthetic perturbations establish fingerprints—copies yield **Utility** collapse, mode collapse degrades **Quality**, corruption and label flips erode **Indistinguishability** or **Quality**. In molecular and image domains we show these patterns reappear: adversarial (WGAN) and flow models echo the low-utility signature, VAEs/flows display fidelity collapse, StyleGAN2 mirrors the sharp-but-narrow profile, and diffusion models achieve balanced performance. Linking stress tests to real generators across the four metrics explains *why* models fail instead of merely ranking them.

## 4.1 Quantile Sensitivity Analysis on Synthetic Data

To test the sensitivity of our quantile-based evaluation strategy—particularly its ability to distinguish fine-grained differences between generators—we design a controlled experimental setting with synthetic data. These experiments do not aim to validate the PAC bounds, but rather to empirically examine how the RankGen metrics, when computed via medians and IQRs, respond to different generative pathologies. We synthesize balanced binary datasets with 27 Gaussian features, split them into surrogate $D_{\text{train1}}$, $D_{\text{train2}}$, $D_{\text{generated}}$, and $D_{\text{test}}$, and apply five perturbations to $D_{\text{generated}}$: (1) label flips, (2) class imbalance, (3) mode collapse to a few real modes, (4) copy noise via near-duplicates from $D_{\text{train1}}$, and (5) additive Gaussian corruption. This controlled suite coaxes distinct failure modes for RankGen to probe (Appendix 13). To test robustness we vary one experimental factor at a time: (i) the number of training examples (100–10k), (ii) the classifier family, (iii) the scoring metric (F1, AUC, precision, recall), and (iv) the positive class ratio. We refer to these as the *dataset size*, *classifier*, *scorer*, and *positive ratio* variations in Table 3. The **Default** configuration corresponds to KNN, F1-score, balanced classes, and 10k samples; all other rows in Table 3 report the Spearman rank correlation between the perturbation magnitude and each metric when we sweep one factor while keeping the others fixed to their default values. We do not aggregate over all sweeps into a single number; instead, we report the per-variation correlations so that the reader can verify that the qualitative fingerprints are stable across experimental choices (see Appendix 14).

Table 3: Spearman rank correlation between metric scores and perturbation magnitude.

| Perturbation | Variation | Quality | Utility | Similarity | Indist. | Exchange. |
|---|---|---|---|---|---|---|
| **Exact Copies** | Dataset Size | 0.1083 | 0.8915 | -0.9923 | -0.9474 | 0.7184 |
| | Classifier | -0.3349 | 0.4261 | -0.9923 | -0.9281 | -0.1549 |
| | Scorer | 0.0469 | 0.9923 | -0.9923 | -0.9735 | 0.9762 |
| | Positive Ratio | -0.3238 | 0.6245 | -0.9922 | -0.9894 | -0.1810 |
| | **Default** | **0** | **1.0000** | **-1.0000** | **-0.9701** | **0.9762** |
| **Mode Collapse** | Dataset Size | 0.7953 | 0.8100 | 0.9923 | 0.9817 | 0.9619 |
| | Classifier | 0.9065 | 0.8844 | 0.9923 | 0.9923 | 0.9896 |
| | Scorer | 0.9659 | 0.9923 | 0.9923 | 0.9923 | 0.9923 |
| | Positive Ratio | 0.7956 | 0.8646 | 0.9898 | 0.9922 | 0.9571 |
| | **Default** | **1.0000** | **1.0000** | **1.0000** | **1.0000** | **1.0000** |
| **Noisy Copies** | Dataset Size | -0.3268 | 0.8438 | -0.9889 | -0.9359 | 0.5627 |
| | Classifier | -0.4106 | 0.3975 | -0.9923 | -0.9105 | -0.1391 |
| | Scorer | -0.0391 | 0.9899 | -0.9923 | -0.9681 | 0.9885 |
| | Positive Ratio | -0.3474 | 0.5376 | -0.9922 | -0.9918 | -0.1851 |
| | **Default** | **0** | **1.0000** | **-1.0000** | **-1.0000** | **1.0000** |
| **Flipping Labels** | Dataset Size | 0.9383 | 0.6937 | 0.9643 | 0.0513 | 0.9494 |
| | Classifier | 0.9880 | 0.7779 | 0.9923 | -0.0866 | 0.9659 |
| | Scorer | 0.9798 | 0.7579 | 0.9923 | 0.2585 | 0.9923 |
| | Positive Ratio | 0.9652 | 0.7371 | 0.9886 | -0.0709 | 0.9852 |
| | **Default** | **1.0000** | **0.7638** | **1.0000** | **0.0127** | **1.0000** |
| **Gaussian Noise** | Dataset Size | 0.9214 | 0.7950 | 0.9735 | 0.5974 | 0.8964 |
| | Classifier | 0.8449 | 0.5967 | 0.9923 | 0.9915 | 0.7219 |
| | Scorer | 0.9923 | 0.8472 | 0.9923 | 0.9897 | 0.9923 |
| | Positive Ratio | 0.9109 | 0.8274 | 0.9087 | 0.9872 | 0.9547 |
| | **Default** | **1.0000** | **0.8729** | **1.0000** | **1.0000** | **1.0000** |

Table 3 shows that each perturbation leaves a compact diagnostic fingerprint: **Utility** alone collapses for exact or noisy copies while **Quality** and **Similarity** stay inflated; mode collapse slashes **Quality** yet scarcely moves **Indistinguishability**; label flips and Gaussian corruption again erode **Quality** but leave **Similarity** high. The quartet is therefore necessary to separate copying, collapse, corruption, and noise.

## 4.2 Real-World Molecular Graph Datasets

We evaluate the quality of molecular graphs generators, a domain where human inspection is infeasible and rigorous metrics are essential. To reflect realistic low-resource settings, models are trained on relatively small datasets (hundreds to a few thousand molecules) without large-scale pretraining. We use a representative subset of classification tasks from the

*Therapeutics Data Commons* (TDS) (Huang et al., 2021): AMES (mutagenicity), BBB (blood–brain barrier penetration), CYP1A2/CYP2C19 (enzyme binding), HERG (cardiotoxicity), and LIPOPHILICITY (logP estimation). Molecules, given as SMILES (Weininger, 1988), are converted into atom–bond graphs. We benchmark several types of generators: **STGG** (Ahn et al., 2021) (autoregressive), **WGAN-GP + R-GCN** (Akensert, 2021) (adversarial), **JTNN** (Jin et al., 2018) and **HierVAE** (Jin et al., 2020) (hierarchical VAEs), **MoFlow** (Zang and Wang, 2020) (flow), **GDSS** (Jo et al., 2022) (score-based diffusion), and **SWINGNN** (Yan et al., 2023) (graph diffusion with shifted windows), **S4DD** (Özçelik et al., 2023)(operates on molecular strings). As a baseline we also introduce a **Neighborhood Swap (NS)** generator, which perturbs input graphs by swapping small ego-subgraphs within instances of the same class iteratively for 1 to 3 iterations (NS-1/2/3). Together these span autoencoding, flow, adversarial, diffusion, and perturbation paradigms (details in App. 16.1, 17.1). Graphs are embedded via (1) fingerprints (**Morgan**, **Torsion**, **Atom Pair**, **RDKit** (O'Boyle et al., 2011)), (2) neural encoders (**GIN** (Xu et al., 2019), **GraphCL** (You et al., 2020), **InfoGraph** (Sun et al., 2019)), and (3) kernels (**NSPDK** (Costa and Grave, 2010)). While overall trends are consistent, rankings can shift across embeddings, underscoring the role of representation in evaluation. More details about each vectorizer in Appendix 15.

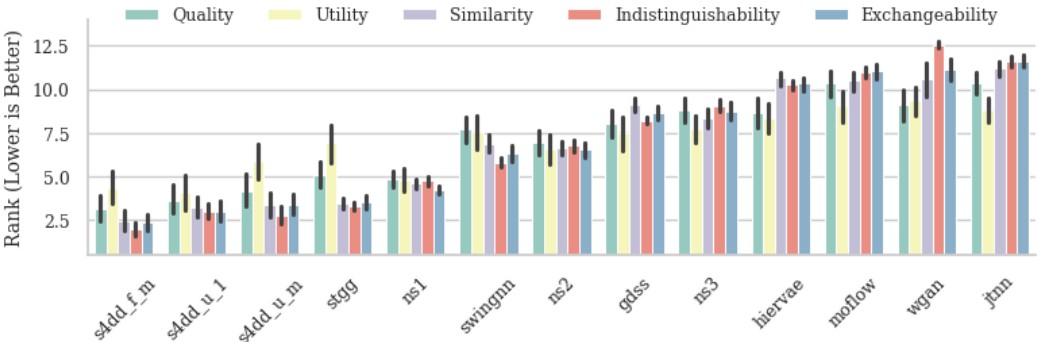

Figure 2: Average generator rank per metric across all TDS datasets and vectorizers (lower is better).

**Results.** Figure 2 shows average quantile-based ranks across datasets and encoders.[1] **S4DD** variants and **STGG** consistently rank highest, reflecting strong generalization and alignment. In contrast, **WGAN** and **MoFlow** perform poorly, especially in **Exchangeability**, which shows that these generators cannot produce molecules useful for predictive tasks. Interestingly, S4DD pretrained on over a million molecules (s4dd_u_m) shows no improvement over dataset-specific training on only a few thousand (s4dd_u_1). The metric breakdown mirrors the synthetic fingerprints: WGAN couples moderate **Indistinguishability** with negligible **Utility**, indicating that its samples offer little downstream value, while flow- and VAE-based models (MoFlow, GDSS, HierVAE, JTNN) exhibit low **Quality** and **Similarity**, consistent with fidelity collapse. By contrast, S4DD and STGG variants remain strong on all four axes, signalling genuinely novel, task-relevant molecules rather than copies.

### 4.3 IMAGE BENCHMARKS: MNIST, FASHION–MNIST, AND CIFAR–10

We use three canonical benchmarks: **MNIST** (LeCun et al., 1998), **Fashion–MNIST** (Xiao et al., 2017), and **CIFAR–10** (Krizhevsky, 2009), with consistent pre-processing and stratified splits (Appendix 22). Test images are never used for training or selection. We evaluate class-conditional generators spanning multiple paradigms. **VAEs:** MLP-VAE, Conv-VAE, and ResNet-VAE (Kingma and Welling, 2014; Rezende et al., 2014).

---

[1]Table 2 reports a single dataset; Fig. 2 averages across all TDS datasets.

**GANs:** DCGAN (Radford et al., 2016), WGAN–GP with projection discriminator (Gulrajani et al., 2017; Miyato and Koyama, 2018), and a 32×32 **StyleGAN2-lite** (Karras et al., 2020). **Diffusion:** cDDPM with UNet backbone (Ho et al., 2020; Ronneberger et al., 2015; Ho and Salimans, 2022) and a Transformer2D backbone (DiT style) (Peebles and Xie, 2023), using HuggingFace Diffusers samplers (von Platen et al., 2022) (see architectural and training details in Appendix 17.2.1). Figure 3 shows RankGen ranks across datasets. RankGen again uncovers characteristic fingerprints. StyleGAN2-lite delivers high **Quality** but almost no **Utility** and weak **Similarity**, mirroring the synthetic mode-collapse profile: crisp yet narrow samples. DCGAN lands near chance in **Utility** while keeping **Indistinguishability** high, signalling shallow realism that fails to expand the task dataset. Diffusion models are the only family balancing all four probes, achieving strong **Exchangeability** by jointly preserving fidelity, task gain, and local mixing. VAEs lag on CIFAR–10 because **Quality** and **Utility** deteriorate, but they remain competitive on MNIST and Fashion–MNIST where capacity demands are lower. Each architecture family is therefore diagnosed with a distinct failure mode instead of being flattened into a single score. The underlying metric values supporting these trends are detailed in Appendix 22.2, and illustrative panels of generated samples are shown in Figures 7, 6, and 5 from Appendix 18. In addition we report in Appendix a direct comparison between RankGen and established metrics such as FID, KID, PRD, and VENDI (Tables 35–43). These results show broad agreement in standard regimes—e.g., diffusion models rank highest across both metric families—while revealing meaningful divergences precisely where classical metrics are known to be unreliable.

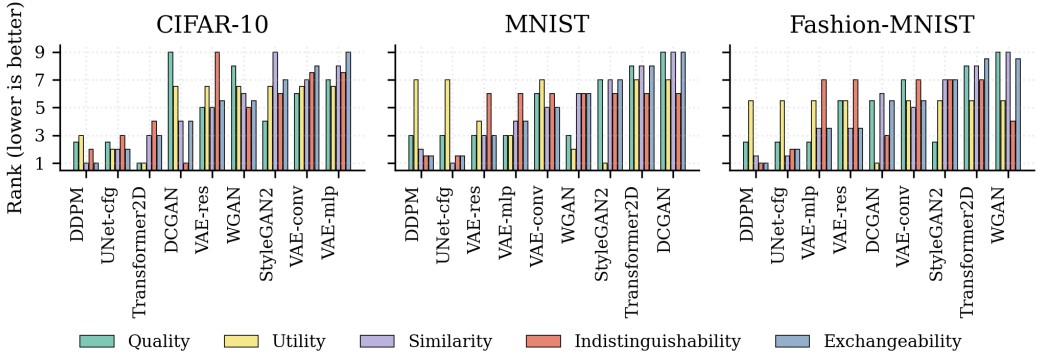

Figure 3: RankGen ranks on **CIFAR–10**, **MNIST**, and **Fashion–MNIST**. Lower is better. StyleGAN2-lite exhibits high Quality but low Utility and Similarity, echoing our synthetic mode-collapse setting. This suggests strong precision but poor coverage—models of this type risk overfitting narrow regions of the data. VAEs remain competitive on digits.

## CONCLUSION

RANKGEN treats evaluation as *diagnosis*: four probes (Quality, Utility, Indistinguishability, Similarity) plus PAC-style bounds and quantile summaries provide reliable, interpretable failure reports, while **Exchangeability** rejects generators that are not simultaneously useful and aligned. Across synthetic, molecular, and image domains RankGen recovers precision–recall trends yet exposes issues scalar metrics miss—copying and corruption signatures, adversarial and flow models' low downstream value, and the contrast between StyleGAN2's sharp-but-narrow samples and diffusion's balanced coverage. It still requires labeled data, task-relevant embeddings, and repeated classifiers, limiting label-scarce settings. Extending RankGen with self-supervised probes and regression tasks is a promising direction. Ultimately RankGen offers a principled alternative to heuristics by revealing *why* generators succeed or fail, enabling safer deployment when human inspection is unfeasible or expensive.

## Acknowledgments

For the purpose of open access, the authors have applied a Creative Commons Attribution (CC BY) license to any Author Accepted Manuscript version arising from this submission.

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

# 5 MODEL RANKING METHODOLOGY

## 5.1 RANKING OF GENERATIVE MODELS

To compare generative models in a principled and uncertainty-aware way, we use a Monte Carlo-based ranking procedure built on four base metrics: *Quality*, *Utility*, *Similarity*, and *Indistinguishability*. Each is computed per dataset and model, then aggregated across samples using quantile-based estimators for the mean and standard deviation.

**Note (summary vs. base metrics).** We also report an *Exchangeability* summary score (Sec. 11), but it is *not* used in the pairwise domination test below to avoid double-counting, since it is derived from the four base metrics.

Let there be $m$ models and $n$ base metrics (here $n = 4$). Each model $M_i$ has a performance vector

$$\boldsymbol{\mu}_i = (\mu_{i1}, \mu_{i2}, \ldots, \mu_{in}), \tag{1}$$

with standard deviations

$$\boldsymbol{\sigma}_i = (\sigma_{i1}, \sigma_{i2}, \ldots, \sigma_{in}), \tag{2}$$

reflecting variability across data splits and training randomness.

**Quantile-to-moment inputs.** The quantities $(\mu_{ik}, \sigma_{ik})$ used below are the robust pseudo-mean and pseudo-standard deviation derived from the empirical quartiles $(Q_1, M, Q_3)$ as described in Sec. 3.8. We use these $(\mu_{ik}, \sigma_{ik})$ only to parameterise the Monte Carlo sampling; the primary reported summaries for each model–metric pair remain the median and IQR. At each Monte Carlo iteration, we draw

$$S_{ik} \sim \mathcal{N}(\mu_{ik}, \sigma_{ik}^2), \quad \text{then clip } S_{ik} \in [0, 1]. \tag{3}$$

Model $M_i$ *dominates* $M_j$ at iteration $t$ iff $S_{ik} \geq S_{jk}$ for all $k$ and $S_{ik} > S_{jk}$ for at least one $k$. We accumulate domination counts $D_{ij}$.

---

**Algorithm 1** Uncertainty-Aware Ranking of Generative Models

---

**Require:** Models $M = \{M_1, \ldots, M_m\}$; means $\boldsymbol{\mu}_i \in [0, 1]^n$; stds $\boldsymbol{\sigma}_i \geq 0$; iterations $N$
**Ensure:** Ranked list of models (using the four base metrics only)
1: Initialize domination count matrix $D \leftarrow \mathbf{0}_{m \times m}$
2: **for** $t = 1$ to $N$ **do**
3:     **for** $i = 1$ to $m$ **do**
4:         **for** $k = 1$ to $n$ **do**
5:             Draw $\tilde{S}_{ik} \sim \mathcal{N}(\mu_{ik}, \sigma_{ik}^2)$; set $S_{ik} \leftarrow \min\{1, \max\{0, \tilde{S}_{ik}\}\}$
6:         **end for**
7:     **end for**
8:     **for** $i = 1$ to $m$ **do**
9:         **for** $j = 1$ to $m$, $i \neq j$ **do**
10:            **if** $S_{ik} \geq S_{jk}$ $\forall k$ and $\exists k : S_{ik} > S_{jk}$ **then**
11:                $D_{ij} \leftarrow D_{ij} + 1$
12:            **end if**
13:         **end for**
14:     **end for**
15: **end for**
16: $R_i \leftarrow \sum_{j=1}^{m} D_{ij}$; sort models by descending $R_i$ and return

---

# 6 Bounds Derivations

Table 4: Glossary of symbols used throughout Secs. 3.3–3.7. A superscript $(i)$ indicates the $i$-th cross-validation split, a **hat** $\widehat{\phantom{x}}$ denotes an empirical estimate from finite data, and a **star** $\star$ denotes the corresponding population (infinite-sample) quantity.

| Symbol | Type | Meaning / definition |
|---|---|---|
| *Datasets and sample sizes* | | |
| $D_{\text{train1}}^{(i)}$ | set | Real training subset used for the baseline classifier. |
| $D_{\text{train2}}^{(i)}$ | | Additional real data used for the upper-bound classifier. |
| $D_{\text{generated}}^{(i)}$ | | Generated data produced by the generator. |
| $D_{\text{test}}$ | | Held-out real test set for *all* performance evaluations. |
| $D_r,\ D_g$ | sets | Real and generated points (context of similarity metric). |
| $D_{\text{mix}} = D_r \cup D_g$ | set | Real+generated pool used to measure local mixing. |
| $n,\ n_r,\ n_g$ | scalars | Total, real and generated sample sizes ($n = n_r + n_g$). |
| $m$ | scalar | Size of the discriminator's validation pool ($m = n_r + n_g$). |
| $k$ | scalar | Number of nearest neighbors in the similarity metric. |
| *Model classes* | | |
| $\Phi_\theta$ | map | Task classifier architecture (fixed across experiments). |
| $\Psi_\theta$ | | Real-vs-generated discriminator. |
| $\Gamma_\theta$ | | Helper for $k$-NN similarity estimation. |
| *Raw classifier accuracies* | | |
| $\rho_1^{(i)}$ | num. | $\Phi_\theta(D_{\text{train1}}^{(i)})$ — baseline real-only. |
| $\rho_2^{(i)}$ | | $\Phi_\theta(D_{\text{generated}}^{(i)})$ — generated-only. |
| $\rho_3^{(i)}$ | | $\Phi_\theta(D_{\text{train1}}^{(i)} \cup D_{\text{train2}}^{(i)})$ — real+real. |
| $\rho_4^{(i)}$ | | $\Phi_\theta(D_{\text{train1}}^{(i)} \cup D_{\text{generated}}^{(i)})$ — real+generated. |
| $\rho_5^{(i)}$ | | $\Psi_\theta$ accuracy (indistinguishability test). |
| $\rho_6^{(i)}$ | | Raw similarity statistic (used to construct $S^{(i)}$). |
| $\rho_j^\star$ | | Population version of the same accuracy. |
| *Derived quantities* | | |
| $\Delta_{\text{real}}$ | num. | $\rho_3^{(i)} - \rho_1^{(i)}$ — gain from more *real* data. |
| $\Delta_{\text{generated}}$ | | $\rho_4^{(i)} - \rho_1^{(i)}$ — gain from *generated* data. |
| $\widehat{\text{quality}}^{(i)}$ | $\dfrac{\rho_2^{(i)}}{\rho_1^{(i)}}$ | Empirical quality on split $i$ (reported clipped to $[0,1]$). |
| $\text{quality}^\star$ | $\dfrac{\rho_2^\star}{\rho_1^\star}$ | Population (infinite-sample) quality. |
| $\widehat{\text{utility}}^{(i)}$ | $\dfrac{\Delta_{\text{generated}}}{\Delta_{\text{real}}}$ | Empirical utility on split $i$. |
| $\text{utility}^\star$ | $\dfrac{\Delta_{\text{generated}}^\star}{\Delta_{\text{real}}^\star}$ | Population utility. |
| $\widehat{\text{indist}}^{(i)}$ | $1 - 2\left|\rho_5^{(i)} - 0.5\right|$ | Empirical indistinguishability on split $i$. |
| $\text{indist}^\star$ | $1 - 2\left|\rho_5^\star - 0.5\right|$ | Population indistinguishability. |
| $S^{(i)}$ | — | Empirical similarity (entropy average) on split $i$. |
| $\text{sim}^\star$ | — | Population similarity. |
| *Error terms and constants in PAC bounds* | | |
| $\varepsilon_{\text{gen}}$ | bnd. | $\sqrt{\log(2/\delta)/(2n)}$ — test-accuracy sampling error. |
| $\varepsilon_{\text{disc}}$ | | $\sqrt{(2d\ln(2m) + \ln(8/\delta))/m}$ — $\mathcal{H}$-discrepancy error. |
| $\varepsilon_{\text{syn}},\ \varepsilon_{\text{real}}$ | | Composite errors for $\Delta_{\text{generated}}$ and $\Delta_{\text{real}}$. |
| $\varepsilon_{\text{sim}}$ | | Error radius for similarity metric. |
| $\mathcal{R}_n$ | scalar | Rademacher complexity of the classifier class. |
| $d$ | scalar | VC dimension of the discriminator's hypothesis class. |
| $L_H$ | scalar | Lipschitz constant of entropy on $[\frac{1}{k+1}, 1 - \frac{1}{k+1}]$ ($\leq 4$). |
| $\lambda$ | num. | Joint optimal risk term in domain discrepancy. |
| $\delta$ | scalar | Confidence level (all PAC bounds hold w.p. $1 - \delta$). |

# 7 BOUND INGREDIENTS AND ASSUMPTIONS

For completeness we collect the constants, estimators, and confidence allocations used in the PAC-style bounds from Sec. 3. This material is supplementary; the main text states only the bound forms.

**Confidence allocation.** Unless otherwise noted, we set $\delta_* = \delta/(MTS)$ where $M$ is the number of models, $T$ the number of metrics, and $S$ the number of resamples. Each bound then holds with probability at least $1 - \delta_*$, and all results jointly with probability at least $1 - \delta$ by a union bound.

**Quality.** We assume $\rho_1^{(i)} > 0$ for the ratio form; if $\rho_1^{(i)} = 0$ we revert to the difference form $\rho_2^{(i)} \geq \rho_1^{(i)} - 2U$. The empirical Rademacher surrogate $\widehat{\mathfrak{R}}$ is computed via a ghost hold-out split.

**Utility.** We estimate $\widehat{\mathrm{disc}}_{\mathcal{H}}$ by training a balanced domain classifier on held-out batches (cross-fitting when necessary), which upper-bounds the $\mathcal{H}\Delta\mathcal{H}$ discrepancy up to constants. We compute $\widehat{\lambda} = \min_{h \in \Phi_\theta} \widehat{R}_{D'_r}(h) + \widehat{R}_{D_s}(h)$ on an auxiliary split disjoint from the evaluation pool. If $\Delta_{\mathrm{real}}^{(i)} < \tau$ with $\tau = 10^{-3}$ we use the difference utility $\Delta_{\mathrm{gen}}^{(i)}$; otherwise we form the ratio $U^{(i)}$.

**Indistinguishability.** The map $g(a) = 1 - 2|a - 0.5|$ from discriminator accuracy $a$ to the indistinguishability score is 2-Lipschitz; our constants absorb this factor. We apply a VC-dimension bound on the discriminator's risk over a balanced pool.

**Similarity.** Neighbourhood label proportions are without-replacement samples. We therefore use Serfling-type deviations for hypergeometric draws. Restricting $p \in [1/(k+1), 1 - 1/(k+1)]$ bounds the entropy derivative $|H'(p)|$, yielding a Lipschitz constant $L_H \leq 4/\ln 2$. Combined with a union bound over $n$ points, this yields the stated $O(\sqrt{\log(n/\delta)/k})$ rate.

**Composite.** We set $\delta_Q = \delta_U = \delta_I = \delta_S = \delta/4$ so that the Exchangeability bound holds with confidence at least $1 - \delta$ by a union bound.

# 8 PAC-STYLE DERIVATION OF THE QUALITY METRIC BOUND

The **quality** score measures how well a classifier trained exclusively on generated data can match the generalization performance of one trained on real data:

$$\mathrm{quality}^{(i)} = \frac{\rho_2^{(i)}}{\rho_1^{(i)}}. \tag{4}$$

Values near 1 indicate that generated data provides almost as much signal as real data (we report $\min\{1, \rho_2^{(i)}/\rho_1^{(i)}\}$ for interpretability).

**Generalization bound.** Let $\ell : \mathcal{X} \times \mathcal{Y} \to \{0, 1\}$ be the 0–1 loss, and define population and empirical risks

$$L(h) = \mathbb{E}_{(x,y) \sim P_{\mathrm{real}}}[\ell(h(x), y)], \qquad \widehat{L}_S(h) = \frac{1}{n} \sum_{(x,y) \in S} \ell(h(x), y). \tag{5}$$

By a standard Rademacher bound Bartlett and Mendelson (2002), with prob. $\geq 1 - \delta$, every $h$ satisfies

$$\left| L(h) - \widehat{L}_S(h) \right| \leq U, \quad U = 2\,\mathcal{R}_n + \sqrt{\frac{\ln(2/\delta)}{2n}}. \tag{6}$$

Let $\widehat{h}_r = \arg\min_h \widehat{L}_{S_r}(h)$ and $\widehat{h}_g = \arg\min_h \widehat{L}_{S_g}(h)$. Then

$$L(\widehat{h}_g) \leq \widehat{L}_{S_g}(\widehat{h}_g) + U \leq \widehat{L}_{S_r}(\widehat{h}_r) + U \leq L(\widehat{h}_r) + 2U. \tag{7}$$

Since $\rho_1^{(i)} = 1 - L(\widehat{h}_r)$ and $\rho_2^{(i)} = 1 - L(\widehat{h}_g)$, we get $\rho_2^{(i)} \geq \rho_1^{(i)} - 2U$. Setting $\varepsilon_1 = 2U = 4\,\mathcal{R}_n + 2\sqrt{\frac{\ln(2/\delta)}{2n}}$ and dividing by $\rho_1^{(i)} > 0$ gives

$$\text{quality}^{(i)} \geq 1 - \frac{\varepsilon_1}{\rho_1^{(i)}}\,. \tag{8}$$

# 9  PAC-style Derivation of the Utility Bound

We consider the lower bound

$$\text{utility}^{(i)} \geq 1 - \frac{\varepsilon_2}{\rho_3^{(i)} - \rho_1^{(i)}} \tag{A.1}$$

quoted in Sec. 3.4. All probabilities below hold with confidence at least $1 - \delta$.

**Notation.**  Let $\mathcal{X} \times \mathcal{Y}$ be the input–label space and $\mathcal{H}$ a hypothesis class of VC-dimension $d$. Define

$$\begin{aligned}
D_{\mathrm{r}} &:= D_{\mathrm{train1}}, & n_{\mathrm{r}} &:= |D_{\mathrm{r}}|, \\
D_{\mathrm{r}}' &:= D_{\mathrm{train2}}, & n_{\mathrm{r}'} &:= |D_{\mathrm{r}}'|, \\
D_{\mathrm{s}} &:= D_{\mathrm{generated}}, & n_{\mathrm{s}} &:= |D_{\mathrm{s}}|, \\
D_{\mathrm{t}} &:= D_{\mathrm{test}}.
\end{aligned}$$

For any sample $S$, let $\hat{h}_S \in \arg\min_{h \in \mathcal{H}} \hat{R}_S(h)$ be the ERM with $\hat{R}_S(h) = \frac{1}{|S|} \sum_{(x,y) \in S} \mathbf{1}\{h(x) \neq y\}$. Let $R(h) = \Pr_{(x,y) \sim D_{\mathrm{t}}}[h(x) \neq y]$.

Define (population) gains from extra real or generated data and their empirical counterparts:

$$\Delta_{\mathrm{real}}^{\star} := R\left(\hat{h}_{D_{\mathrm{r}}}\right) - R\left(\hat{h}_{D_{\mathrm{r}} \cup D_{\mathrm{r}}'}\right), \quad \Delta_{\mathrm{syn}}^{\star} := R\left(\hat{h}_{D_{\mathrm{r}}}\right) - R\left(\hat{h}_{D_{\mathrm{r}} \cup D_{\mathrm{s}}}\right), \tag{A.2}$$

with $\hat{\Delta}_{\mathrm{real}} = \rho_3^{(i)} - \rho_1^{(i)}$ and $\hat{\Delta}_{\mathrm{syn}} = \rho_4^{(i)} - \rho_1^{(i)}$.

**Step 1: domain-adaptation inequality.**  Following Ben-David et al. (2010), for any $h$ and distributions $P, Q$,

$$|R_Q(h) - R_P(h)| \leq \mathrm{disc}_{\mathcal{H}}(P, Q) + \lambda(P, Q), \tag{A.3}$$

where $\mathrm{disc}_{\mathcal{H}}$ is the $\mathcal{H}$-discrepancy and $\lambda$ the joint optimal risk. Apply (A.3) to $P = D_{\mathrm{r}}'$, $Q = D_{\mathrm{s}}$ with $h = \hat{h}_{D_{\mathrm{r}} \cup D_{\mathrm{s}}}$ to obtain

$$R_{\mathrm{s}} - R_{\mathrm{r}}' \leq \mathrm{disc}_{\mathcal{H}}(D_{\mathrm{r}}', D_{\mathrm{s}}) + \lambda. \tag{A.4}$$

Combining with (A.2) yields

$$\Delta_{\mathrm{syn}}^{\star} \geq \Delta_{\mathrm{real}}^{\star} - \left[\mathrm{disc}_{\mathcal{H}}(D_{\mathrm{r}}', D_{\mathrm{s}}) + \lambda\right]. \tag{A.5}$$

**Step 2: empirical–population deviations.**  With probability $\geq 1 - \frac{\delta}{2}$ (after apportioning $\delta$ over models and resampling splits),

$$\left|R_{\mathrm{r}} - \hat{R}_{\mathrm{r}}\right|, \left|R_{\mathrm{r}}' - \hat{R}_{\mathrm{r}}'\right|, \left|R_{\mathrm{s}} - \hat{R}_{\mathrm{s}}\right| \leq \varepsilon_{\mathrm{gen}}, \qquad \varepsilon_{\mathrm{gen}} = \sqrt{\frac{\ln(2/\delta)}{2\,n_{\min}}}, \tag{A.6}$$

$$\mathrm{disc}_{\mathcal{H}}(D_{\mathrm{r}}', D_{\mathrm{s}}) \leq \widehat{\mathrm{disc}}_{\mathcal{H}}(D_{\mathrm{r}}', D_{\mathrm{s}}) + \varepsilon_{\mathrm{disc}}, \quad \varepsilon_{\mathrm{disc}} = \sqrt{\frac{2d\ln(2m) + \ln(8/\delta)}{m}}, \tag{A.7}$$

where $n_{\min} := \min\{n_{\mathrm{r}}, n_{\mathrm{r}}', n_{\mathrm{s}}\}$ and $m = n_{\mathrm{r}'} + n_{\mathrm{s}}$.

Plugging into (A.5) and rearranging, with probability $\geq 1 - \frac{\delta}{2}$,

$$\hat{\Delta}_{\mathrm{syn}} \geq \hat{\Delta}_{\mathrm{real}} - \left[\widehat{\mathrm{disc}}_{\mathcal{H}}(D_{\mathrm{r}}', D_{\mathrm{s}}) + \varepsilon_{\mathrm{disc}} + \lambda + 2\varepsilon_{\mathrm{gen}}\right]. \tag{A.8}$$

**Step 3: the utility ratio.** If $\hat{\Delta}_{\mathrm{real}} > 0$ (else set $\mathrm{utility}^{(i)} = 0$), divide (A.8) by $\hat{\Delta}_{\mathrm{real}}$ to obtain

$$\mathrm{utility}^{(i)} = \frac{\hat{\Delta}_{\mathrm{syn}}}{\hat{\Delta}_{\mathrm{real}}} \geq 1 - \frac{\widehat{\mathrm{disc}}_{\mathcal{H}}(D'_{\mathrm{r}}, D_{\mathrm{s}}) + \varepsilon_{\mathrm{disc}} + \lambda + 2\varepsilon_{\mathrm{gen}}}{\hat{\Delta}_{\mathrm{real}}}. \tag{A.9}$$

**Step 4: notation match.** We estimate the joint optimal risk via

$$\widehat{\lambda} := \min_{h \in \mathcal{H}} \widehat{R}_{D'_{\mathrm{r}}}(h) + \widehat{R}_{D_{\mathrm{s}}}(h), \tag{9}$$

using an auxiliary split (or cross-fitting) that remains disjoint from the evaluation data. Define

$$\varepsilon_2 := \underbrace{\widehat{\mathrm{disc}}_{\mathcal{H}}(D'_{\mathrm{r}}, D_{\mathrm{s}})}_{\text{observed}} + \underbrace{\varepsilon_{\mathrm{disc}}}_{\varepsilon_3} + \underbrace{2\varepsilon_{\mathrm{gen}}}_{\varepsilon_4^{\mathrm{util}}} + \widehat{\lambda}, \tag{A.10}$$

so (A.9) matches (A.1). In practice we guard against small denominators by reporting the ratio bound only when $\hat{\Delta}_{\mathrm{real}} > \tau$ (Section 3.4). Otherwise we fall back to the difference guarantee $\hat{\Delta}_{\mathrm{syn}} \geq \hat{\Delta}_{\mathrm{real}} - (\widehat{\mathrm{disc}}_{\mathcal{H}} + \varepsilon_{\mathrm{disc}} + \widehat{\lambda} + 2\varepsilon_{\mathrm{gen}})$.

## 10  PAC-style Derivation of the Similarity Bound

Throughout this appendix we write $S^{(i)}$ for the empirical similarity score defined in Section 3.6, and $S^{\star}$ for its population counterpart.

### 10.1  Problem Definition

Let $D_r = \{X_r, y_r\}$ and $D_g = \{X_g, y_g\}$ be real and generated datasets. Merge to $X = X_r \cup X_g$ and define binary origin labels $Y \in \{0, 1\}$.

### 10.2  Similarity Computation

Use cosine similarity $K(x, x') = \frac{x^{\top} x'}{\|x\| \, \|x'\|}$ to form $k$-NN neighborhoods $\mathcal{N}(x_i)$. Let $p_i$ be the in-domain fraction among neighbors of $x_i$ and

$$H_i = -p_i \log p_i - (1 - p_i) \log(1 - p_i), \qquad S^{(i)} = \frac{1}{|X|} \sum_i H_i. \tag{10}$$

### 10.3  Derivation of the Similarity Bound

Let $Z_x = k \hat{p}_x$ count in-domain neighbours. Because we sample without replacement from $D_r \cup D_g$, $Z_x$ follows a hypergeometric law; Serfling's inequality yields

$$\Pr\left[ |\hat{p}_x - p_x| > \sqrt{\tfrac{\log(2/\delta_1)}{2k}} \right] \leq \delta_1. \tag{B.2}$$

On $p \in [\tau, 1 - \tau]$ with $\tau = \frac{1}{k+1}$, entropy $H$ is $L_H$-Lipschitz with $L_H \leq 4/\ln 2$, so with prob. $\geq 1 - \frac{\delta}{2}$ (allocating $\delta/(2n)$ per point and union-bounding),

$$|H(\hat{p}_x) - H(p_x)| \leq L_H \sqrt{\tfrac{\log(2n/\delta)}{2k}} \quad \forall x. \tag{B.3}$$

Averaging and applying Hoeffding to the empirical mean gives

$$\left| \tfrac{1}{n} \sum_x H(p_x) - \tfrac{1}{n} \sum_x H(\hat{p}_x) \right| \leq \sqrt{\tfrac{\log(2/\delta)}{2n}} \log 2. \tag{B.4}$$

Combining (B.3)–(B.4) gives, with probability at least $1 - \delta$,

$$|S^{(i)} - S^{\star}| \leq \frac{1}{\log 2} \left( L_H \sqrt{\frac{\log(2n/\delta)}{2k}} + \log 2 \sqrt{\frac{\log(2/\delta)}{2n}} \right).$$

Equivalently, since $S^\star \leq 1$, this deviation bound implies the lower bound

$$S^\star \geq 1 - \frac{\varepsilon_{\mathrm{sim}}}{\log 2},$$

which appears as Eq. (B.7).

## 11 PAC-style Derivation of the Exchangeability Metric Bound

Define $\Delta_{\mathrm{qual}} = (Q + U)/2$ and $\Delta_{\mathrm{sim}} = (I + S)/2$. The *exchangeability score* is

$$\mathcal{E}_{\mathrm{min}} = \min\{\Delta_{\mathrm{qual}}, \Delta_{\mathrm{sim}}\}. \tag{12}$$

If $Q \geq 1 - \varepsilon_Q$, $U \geq 1 - \varepsilon_U$, $I \geq 1 - \varepsilon_I$, $S \geq 1 - \varepsilon_S$, then

$$\mathcal{E}_{\mathrm{min}} \geq 1 - \max\left\{\frac{\varepsilon_Q + \varepsilon_U}{2}, \frac{\varepsilon_I + \varepsilon_S}{2}\right\}. \tag{13}$$

## 12 Default Task Classifiers

RankGen relies on a downstream task classifier to compute all four metrics $(\rho_1^{(i)}, \ldots, \rho_6^{(i)})$. To avoid confounding generator differences with classifier changes, we use fixed default classifiers in all experiments, except where explicitly overridden.

**Real-data experiments (molecules and images).** For all molecular and image benchmarks we use a single, fixed task classifier:

- **Classifier:** ExtraTreesClassifier (scikit-learn)
- **Number of trees:** 300
- **Parallelization:** n_jobs = -1
- **Criterion:** Gini impurity
- **Bootstrap:** False
- **Max features:** "auto"
- **Scoring: F1 score** on the held-out test set

This classifier is used for all RankGen metrics across all real-data experiments to ensure a consistent, architecture-independent assessment.

**Synthetic sanity-check experiments.** For the synthetic perturbation experiments (Sec. 4.1), the *default* task classifier is:

- **Classifier:** $k$-Nearest Neighbors (KNN)
- **Distance metric:** Euclidean
- **Number of neighbors:** $k = 5$
- **Scoring: F1 score**

## 13 Perturbation Techniques Applied to the Synthetically Generated Dataset

In this study, we applied several perturbation techniques to the generated dataset to simulate real-world challenges such as label noise, class imbalance, and data corruption. These perturbations are designed to test the robustness of the rank given by each metric in handling imperfect data.

All experiments were designed to start with $D_{\mathrm{generated}} \approx D_{\mathrm{train}}$, and apply a perturbation degree $t \in [0, 1]$ that quantifies the dissimilarity between the generated and reference

datasets. To objectively assess the performance of each metric, we compute the Spearman rank correlation between the metric scores $\hat{\rho}$ and the perturbation degree $t$. An ideal metric would exhibit a rank correlation of 1.0, indicating a perfect monotonic relationship between the perturbation degree and the metric score.

We applied the following perturbation techniques:

- **Mode Collapse:** For each class, we drop one or more K-means subclusters and re-assign their points to the remaining submodes, producing reduced diversity without altering class counts.

- **Class Imbalance:** Introduced by either increasing or decreasing the number of instances in each class, based on a specified ratio or exact count.

- **Label Noise:** Added by randomly flipping a fraction of the target labels in the dataset.

- **Exact Copy Noise:** Introduced by replacing a subset of the $D_{\text{generated}}$ instances with copies of samples drawn from $D_{\text{train}}$.

- **Noise Copies:** A subset of $D_{\text{generated}}$ is replaced with near-duplicates of real training samples from $D_{\text{train}}$, each perturbed by zero-mean Gaussian noise with standard deviation $\sigma = 0.05$.

- **Gaussian Noise:** Both additive and multiplicative noise were applied to the instances, and features were swapped to simulate data corruption.

## 14 Classifiers, Dataset Sizes, and Scorers Used in the synthetic Experiments

We evaluate multiple dataset sizes (100–10000), classifiers (e.g., ExtraTrees, RandomForest, Logistic Regression, SVC, KNN), and scorers.

| Dataset Size | Classifier | Scorer | Positive Class Ratio |
|---|---|---|---|
| 100 | ExtraTreesClassifier | F1 Score | 0.40 |
| 500 | RandomForestClassifier | Balanced Accuracy | 0.50 |
| 1000 | GradientBoostingClassifier | Precision | 0.20 |
| 2500 | AdaBoostClassifier | Recall | 0.60 |
| 5000 | LogisticRegression | ROC-AUC Score | 0.35 |
| 10000 | DecisionTreeClassifier | Jaccard Score | 0.25 |
| | SVC | Average Precision | 0.90 |
| | KNeighborsClassifier | Precision | 0.30 |
| | GaussianNB | | 0.80 |
| | ExtraTreesClassifier | | 0.95 |
| | RandomForestClassifier | | 0.15 |
| | GradientBoostingClassifier | | 0.10 |

### 14.1 Experimental Setup and Function Parameters

| Parameter | Value | Description |
|---|---|---|
| n_iterations | 100 | Number of iterations for scoring computation. |
| use_resampling | True | Whether to resample the data during evaluation. |
| use_replacement | False | Whether to sample with replacement. |
| fraction | 0.8 | Fraction of data to use for training when resampling. |
| data_estimator | KNeighborsClassifier | Classifier used to estimate model performance. |
| scorer | F1, AUC, etc. | Scoring functions used to evaluate discriminative performance. |
| verbose | 2 | Level of verbosity for logging during scoring. |
| parallel | True | Whether computations are parallelized for efficiency. |

## 15 Vectorizers Used in the Graph Data Experiments

In this study, we utilized several vectorizers for our experiments. Below is a brief overview of each method used:

- GIN (Xu et al., 2019) Xu et al. (2019): The Graph Isomorphism Network (GIN) is a graph neural network based on the idea of learning a graph's structural properties through a series of message-passing layers. GIN has demonstrated strong performance on graph classification tasks, especially when combined with sufficient training data and appropriate network architectures.

- GraphCL (You et al., 2020) You et al. (2020): GraphCL is a contrastive learning method for graph data, whcih leverages a self-supervised learning framework that maximizes the agreement between augmented views of a graph while minimizing the distance between distinct views. This technique helps capture better graph representations, even without labeled data, making it suitable for semi-supervised learning settings.

- InfoGraph (Sun et al., 2020) Sun et al. (2019): InfoGraph is a method for unsupervised and semi-supervised graph learning that focuses on learning graph-level representations. It uses mutual information maximization to capture global graph structures, enabling the model to generate meaningful and informative embeddings.

- GraphVectorizer (Custom Implementation): The GraphVectorizer processes graph structures derived from SMILES strings. It extracts few graph features, including degree histograms, clustering coefficients, and node labels. These features are then encoded into fixed-length vectors, capturing graph topology and node-level information. The vectorizer includes padding to standardize feature lengths and applies label encoding to handle node labels.

-NSPDK (Costa and Grave, 2010) Costa and Grave (2010): The Fast Neighborhood Subgraph Pairwise Distance Kernel (NSPDK) computes the similarity between two graphs based on the pairwise distances between their corresponding subgraphs. This kernel is designed to efficiently capture structural information by evaluating local neighborhoods and pairwise distances.

- HashedAPVectorizer O'Boyle et al. (2011): Computes Atom Pair fingerprints by capturing relationships between atom pairs in a molecule, resulting in a fixed-length hashed vector. This vectorizer is used for molecular similarity tasks.

- HashedMorganVectorizer O'Boyle et al. (2011): Generates Morgan fingerprints, which are circular and capture local atom environments. It is widely used in molecular similarity and classification tasks.

- HashedRDKVectorizer O'Boyle et al. (2011): Computes RDKit Fingerprints, which are based on atom connectivity and properties. This vectorizer is suitable for general molecular similarity and topological feature representation.

- HashedTorsionVectorizer O'Boyle et al. (2011): Generates Topological Torsion fingerprints, focusing on torsion angles between atoms to capture 3D molecular arrangements. This vectorizer is useful for tasks that involve molecular conformation analysis.

## 16 Datasets

### 16.1 Molecular Datasets

We conduct experiments using several datasets from the Therapeutic Data Commons Huang et al. (2021), which are designed for single prediction tasks. By default, the datasets are split into training, testing, and validation sets with proportions of 70%, 20%, and 10%, respectively, using random splitting. The raw data is initially provided as SMILES strings Weininger (1988). For this study, we apply multiple vectorization techniques, including both *molecular* and *graph-based* vectorizers. The molecular vectorizers process the SMILES strings and generates molecular features, which are then used by our metric. Additionally,

we employ the graph-based vectorizers to convert SMILES into graph representations, where nodes correspond to atoms and edges correspond to bonds in the molecule. Each record is represented as a graph $G = (V, E, X, \epsilon)$, where $V$ represents the vertices (atoms), $E$ represents the edges (bonds), with $E = \{(i,j) \mid i,j \in \{1, \ldots, |V|\}\}$. $X \in \mathbb{R}^{N \times D}$ is the node feature matrix, $E \in \mathbb{R}^{(N \times N \times 1)}$ is the edge attribute matrix, and $N$ is the number of nodes. The dimensionality of the node features $D$ is set to 9, and the dimensionality of the edge features $F$ is set to 3. The node features for molecular graph generation include: atomic number, chirality, degree, formal charge, number of hydrogen atoms, number of radical electrons, hybridization, aromaticity, and whether the atom is part of a ring. The edge features include: bond type, bond stereo, and whether the bond is conjugated.PyTorch Geometric Contributors (2021)

- **AMES Mutagenicity**: Consists of 7,255 binary drugs labeled as mutagenic or non-mutagenic with $3 \leq |V| \leq 55$.

- **BBB Martins**: 1,975 drugs classified into those that can penetrate the blood-brain barrier (BBB) or not with $2 < |V| < 132$.

- **Cyp1a2 Veith**: 12,579 drugs classified into those that can / cannot inhibit the CYP1A2 gene with $7 \leq |V| \leq 123$.

- **Cyp2c19 Veith**: 12,665 drugs classified into ones that can / cannot inhibit the CYP2C19 gene with $7 \leq |V| \leq 114$.

- **Herg Karim**: Consists of 13,445 molecular structures labeled as hERG ($< 10\,\mu M$) and non-hERG ($\geq 10\,\mu M$) blockers with $6 < |V| < 58$.

- **Lipophilicity AstraZeneca**: 4,200 binary labeled drugs based on their ability to dissolve in lipids.

Table 7: Sizes of training and testing sets used in our experiments with real generators.

| Dataset | $D_{\mathbf{gen}}$ | $D_{\mathbf{train1}}$ | $D_{\mathbf{train2}}$ | $D_{\mathbf{test}}$ |
|---|---|---|---|---|
| ames | 2340 | 2340 | 2340 | 1316 |
| bbb_martins | 318 | 318 | 318 | 206 |
| cyp1a2_veith | 4128 | 4401 | 4128 | 2376 |
| cyp2c19_veith | 3956 | 3956 | 3956 | 2326 |
| herg_karim | 4700 | 4700 | 4700 | 2686 |
| lipophilicity_astrazeneca | 524 | 524 | 524 | 316 |

Table 8: General statistics for molecular datasets used in our experiments. Node types are shown for positive and negative samples separately.

| Dataset | Size | Positives | Negatives | Node Types (Pos, Neg) | Edge Types |
|---|---|---|---|---|---|
| ames | 5094 | 2759 | 2335 | 10 (Pos), 10 (Neg) | 4 |
| bbb_martins | 1421 | 1096 | 325 | 12 (Pos), 11 (Neg) | 4 |
| cyp1a2_veith | 8805 | 4060 | 4745 | 17 (Pos), 25 (Neg) | 4 |
| cyp2c19_veith | 8866 | 4063 | 4803 | 17 (Pos), 25 (Neg) | 4 |
| herg_karim | 9412 | 4714 | 4698 | 10 (Pos), 15 (Neg) | 4 |
| lipophilicity_astrazeneca | 2940 | 2446 | 494 | 12 (Pos), 10 (Neg) | 4 |

## 16.2 SYNTHETIC DATASET

The generated dataset in this study was generated using scikit-learn's **make_classification** function, producing 27 features sampled from Gaussian distributions. Initially the dataset is balanced, with equal class distributions (50% positive and 50% negative), and includes both informative and redundant features. The data is split into four sets—training, reference, test, and generated—using stratified sampling to maintain class balance. While the features are Gaussian by default, they can be adapted for non-Gaussian experiments.

Table 9: Atomic Numbers for Positive and Negative Classes

| Atomic Numbers (Positive) |
|---|
| $[1, 35, 6, 7, 8, 9, 15, 16, 17, 53]$ |
| $[1, 5, 6, 7, 8, 9, 11, 15, 16, 17, 35, 53]$ |
| $[1, 3, 6, 7, 8, 9, 11, 14, 15, 16, 17, 80, 78, 20, 24, 25, 26, 29, 30, 33, 34, 35, 50, 51, 53]$ |
| $[1, 5, 6, 7, 8, 9, 11, 14, 15, 16, 17, 80, 19, 78, 26, 29, 35, 53]$ |
| $[1, 35, 6, 7, 8, 9, 14, 16, 17, 53]$ |
| $[1, 6, 7, 8, 9, 14, 15, 16, 17, 34, 35, 53]$ |
| **Atomic Numbers (Negative)** |
| $[1, 35, 6, 7, 8, 9, 15, 16, 17, 53]$ |
| $[1, 35, 6, 7, 8, 9, 11, 15, 16, 17, 53]$ |
| $[1, 5, 6, 7, 8, 9, 11, 78, 15, 16, 17, 14, 19, 20, 25, 26, 29, 30, 33, 34, 35, 44, 50, 51, 53]$ |
| $[1, 5, 6, 7, 8, 9, 11, 78, 15, 16, 17, 14, 19, 20, 80, 25, 26, 29, 30, 33, 34, 35, 44, 50, 51, 53]$ |
| $[1, 5, 6, 7, 8, 9, 11, 14, 15, 16, 17, 79, 34, 35, 53]$ |
| $[1, 35, 5, 6, 7, 8, 9, 15, 16, 17]$ |

## 16.3 IMAGE DATASETS

We evaluate on three canonical benchmarks: **MNIST** (LeCun et al., 1998), **Fashion–MNIST** (Xiao et al., 2017), and **CIFAR–10** (Krizhevsky, 2009). All models share preprocessing and canonical stratified splits (details in Appendix 22); test images are never used for training or model selection.

## 17 GENERATORS USED

### 17.1 MOLECULAR(GRAPH) DATA GENERATION

In this study, we utilized several graph generation models for our experiments. Below is a brief overview of each method used, along with the corresponding citations:

Table 10: Overview of Generative Models Used in This Study

| Generator | Model Class | Description |
|---|---|---|
| STGG Ahn et al. (2021) | Autoregressive | Spanning-tree-based decoder with Transformer backbone |
| WGAN-GP Akensert (2021) | GAN-based | Adversarial model using R-GCN for graph discrimination |
| JTNN Jin et al. (2018) | VAE-based | Junction-tree structured variational autoencoder |
| HierVAE Jin et al. (2020) | VAE-based | Hierarchical VAE with coarse-to-fine motif decoding |
| MoFlow Zang and Wang (2020) | Flow-based | Conditional normalizing flow model for molecules |
| GDSS Jo et al. (2022) | Diffusion-based | SDE-based generative diffusion over graphs |
| S4DD Özçelik et al. (2024) | State Space / Hybrid | Dual-mode state-space model with Transformer-style decoding |
| SWINGNN Yan et al. (2023) | Diffusion-based | 2-WL guided diffusion with shifted-window attention |
| Neighborhood Swap (NS-1/2/3) | Perturbation-based | Iterative, non-parametric local rewiring for augmentation |

- **STGG (Spanning Tree Graph Generator)** Ahn et al. (2021): Formulates the graph generation of a molecular graph as a sequence of tree-constructive operations applied through the composition of a spanning tree with the residual edges. STGG adopts a transformer architecture which generates the tree by using relative positional encodings and residual edges using an attention-based predictor.

- **WGAN-GP with R-GCN (Wasserstein GAN with Gradient Penalty and Relational Graph Convolutional Networks)** Akensert (2021): Originally designed for the generation of small molecular graphs such as QM9, but adjusted to generate larger compounds. The generator network consists of two fully connected networks, and the discriminator implements non-linearly transformed neighborhood aggregations through the means of R-GCN.

- **JTNN (Junction Tree Variational Autoencoder for Molecular Graph Generation)** Jin et al. (2018): Generates graphs in two stages. In the first stage, a tree structured object is generated by exploiting the coarse-grained representations of the training molecular graphs. In the second stage, the nodes in the tree (which are essentially subgraphs) are assembled back into molecules.

- **HierVAE** Jin et al. (2020): Uses a hierarchical encoder-decoder architecture for graph generation. The encoder produces a multi-resolution representation for each input graph in a fine to coarse fashion, starting with the atoms and finishing with fully connected motifs. During the decoding process, the molecules are assembled back in a coarse to fine manner, where three consecutive predictions are made at each pass: new motif selection, which part of it attaches, and the points of contact with the current molecule.

- **Moflow** Zang and Wang (2020): Works by using two different graph conditional flows, one for atoms and one for bonds, for obtaining their latent representations. The molecule generation uses the inverse transformations of the inference operations, followed by post-validity correction.

- **GDSS** Jo et al. (2022): Proposes a continuous-time SDE system to model the diffusion process over nodes and edges simultaneously, where Gaussian noise is directly inserted into the adjacency matrix and node features. Sampling is done by solving the SDE used to describe the reverse-time diffusion process.

- **S4DD** or S4 for Denovo Molecule Design as in Özçelik et al. (2024):A dual-mode generative model that combines the recurrence of LSTMs with the global context modeling of Transformers, trained via global convolution and recurrent sequence generation. We evaluate three training configurations, each followed by fine-tuning on our target datasets:

  - `unfiltered_1`: trained directly on the target dataset without pretraining or filtering.

  - `unfiltered_m`: pretrained on a large, diverse set of over 1 million molecules (e.g., toxic and non-toxic compounds from the Therapeutics Data Commons), then fine-tuned.

  - `filtered_m`: pretrained on a combined dataset (ZINC + QM9 + ChEMBL), filtered to include only common atom types, then fine-tuned.

- **SWINGNN** Yan et al. (2023): A non-invariant diffusion model that employs an edge-to-edge 2-WL message passing network and utilizes shifted window-based self-attention. They propose a 2nd order sampler with correction for generating large molecular graphs.

- **Neighborhood Swap Generator (NS)**: A non-parametric, perturbation-based generator that produces augmented graphs through iterative local rewiring. It begins by decomposing each input graph into small neighborhoods (e.g., 1–2-hop subgraphs) and then perturbs them via controlled rewiring and recombination. The generator operates in multiple sequential rounds, where each round applies perturbations not only to the original structure but also to neighborhoods altered in previous steps. This compounding process introduces increasingly diverse graph structures while preserving essential properties. We define three versions—NS-1, NS-2, and NS-3—corresponding to one, two, and three rounds of perturbation, respectively. NSG requires no training and is well-suited for data augmentation and robustness evaluation in graph learning tasks.

### 17.1.1 Hyperparameters for molecular graph generators

Table 11: HierVAE Hyperparameters

| Parameter | Value | Parameter | Value |
|---|---|---|---|
| rnn_type | LSTM | depthG | 15 |
| hidden_size | 250 | diterT | 1 |
| embed_size | 250 | diterG | 3 |
| batch_size | 50 | dropout | 0 |
| latent_size | 32 | learning_rate | $1 \times 10^{-3}$ |
| depthT | 15 | clip_norm | 5 |
| step_beta | $1 \times 10^{-3}$ | max_beta | 1 |
| warmup | 10000 | kl_anneal_rate | 0.9 |
| epochs | 25000 | – | – |

Table 12: JTNN Hyperparameters

| Parameter | Value | Parameter | Value |
|---|---|---|---|
| epochs | 5 | beta | 0 |
| learning_rate | $1 \times 10^{-3}$ | max_beta | 1 |
| annealing_rate | 0.9 | save_iter | 5000 |
| clip_norm | 50 | step_beta | 0.002 |
| total_step | 0 | annealing_iterations | 40000 |
| kl_anneal_iter | 2000 | hidden_size | 56 |
| latent_size | 40 | – | – |

Table 13: S4forDenovoDesign S4DD Hyperparameters

| Parameter | Value | Parameter | Value |
|---|---|---|---|
| model_dim | 256 | state_dim | 64 |
| n_layers | 4 | n_ssm | 1 |
| dropout | 0.25 | vocab_size | 50 |
| sequence_length | 120 | n_max_epochs | 400 |
| learning_rate | $1 \times 10^{-3}$ | batch_size | 500 |
| device | "cuda" | – | – |

Table 14: GDSS Model Hyperparameters

| Parameter | Value | Parameter | Value |
|---|---|---|---|
| s_theta | 2 | s_phi | 16 |
| GCN_layers | 4 | hidden_dim | 2 |
| attention_heads | 8 | initial_channels | 4 |
| hidden_channels | 3 | final_channels | 16 |
| SDE_X | VP | sampling_steps_X | 1000 |
| $\beta_{\min}^{(X)}$ | 0.1 | $\beta_{\max}^{(X)}$ | 1 |
| SDE_A | VE | sampling_steps_A | 1000 |
| $\beta_{\min}^{(A)}$ | 0.2 | $\beta_{\max}^{(A)}$ | 1 |
| solver | Rev + Langevin | – | – |

Table 15: GDSS Training Settings

| Setting | Optimizer | LR | Weight Decay | Batch | Epochs | EMA |
|---|---|---|---|---|---|---|
| Mol Graphs | Adam | $5 \times 10^{-3}$ | $1 \times 10^{-4}$ | 56 | 1000 | 0.999 |

Table 16: STGG Hyperparameters

| LR | Optimizer | Epochs | Layers | Embedding Size | Dropout |
|---|---|---|---|---|---|
| 0.0001 | Adam | 100 | 3 | 1024 | 0.1 |

Table 17: MoFlow Hyperparameters

| Batch | LR | LR Decay | Epochs | `b_n_flow` | `b_n_block` | `b_hidden_ch` |
|---|---|---|---|---|---|---|
| 12 | 0.001 | 0.999995 | 200 | 10 | 1 | 128/128 |

| `b_conv_lu` | `a_n_flow` | `a_n_block` | `a_hidden_gnn` | `a_hidden_lin` | `learn_dist` | `noise_scale` |
|---|---|---|---|---|---|---|
| 1 | 27 | 1 | 64 | 128/64 | True | 0.6 |

Table 18: Neighborhood Swap Generator Hyperparameters

| Version | `num_iterations` | `min_size` | `max_size` | `nbits` | `size` | `max_permutations` | `parallel` |
|---|---|---|---|---|---|---|---|
| NS1 | 1 | 1 | 2 | 12 | 20 | 2 | True |
| NS2 | 2 | 1 | 2 | 12 | 20 | 2 | True |
| NS3 | 3 | 1 | 2 | 12 | 20 | 2 | True |

Table 19: WGAN-GP Hyperparameters

| Parameter | Value |
|---|---|
| Dense Units | 128, 256, 512 |
| Dropout Rate | 0.2 |
| Discriminator Steps | 1 |
| GConv Units | 128, 128, 128, 128 |
| Generator Dense Units | 512, 512 |
| Generator Dropout Rate | 0.2 |
| Generator Steps | 1 |
| Gradient Penalty Weight | 10 |
| Optimizer | Adam |
| Learning Rate | 0.001 |
| Epochs | 20 |
| Batch Size | 28 |

Table 20: SWINGNN Hyperparameters

| Graph Type | MCMC | Model | Precond | Sigma Dist | Steps |
|---|---|---|---|---|---|
| Molecular | edm | edm | edm | 256 | -1, 1, $x_0$ |

| Model Name | Feature Dims | Depths | Window Size | Patch Size | Sample Clip |
|---|---|---|---|---|---|
| swin_gnn | Molecular | 1 1 3 1 | 2 | 3 | – |

## 17.2 Image data Generators

### 17.2.1 Generative Models and Training Setups

We evaluate conditional VAEs (Kingma and Welling, 2014; Rezende et al., 2014), GAN variants (DCGAN (Radford et al., 2016), WGAN-GP with projection critic (Gulrajani et al., 2017; Miyato and Koyama, 2018), StyleGAN2-lite adapted to 32×32 (Karras et al., 2020)), and conditional diffusion models (UNet backbones (Ronneberger et al., 2015) and

Transformer2D/DiT-style backbones (Peebles and Xie, 2023)) trained as DDPMs (Ho et al., 2020) with classifier-free guidance.

**Conditional VAEs.** Shared training hyperparameters (per dataset) and compact architecture deltas are listed in Tables 21 and 22. All VAEs are class-conditional with label embeddings concatenated at encoder/latent/decoder as appropriate.

Table 21: VAE training setup shared across MLP/Conv/Res (per dataset).

| Dataset | $z$ | Cond | Ep | Bs | LR | KW | $\beta$ | FB | Loss | Aug |
|---|---|---|---|---|---|---|---|---|---|---|
| MNIST | 32 | 32 | 80 | 128 | $1 \times 10^{-3}$ | 20 | 1.0 | 0.0 | BCE | none |
| FashionMNIST | 32 | 32 | 100 | 128 | $1 \times 10^{-3}$ | 25 | 1.0 | 0.0 | BCE | none |
| CIFAR-10 | 128 | 64 | 250 | 128 | $1 \times 10^{-3}$ | 30 | 1.0 | 0.0 | MSE | flip + crop |

Table 22: VAE architecture details; decoders mirror encoders unless noted.

| Arch | Datasets | Encoder | Decoder / Notes |
|---|---|---|---|
| MLP | MNIST / FashionMNIST | FC: 512, 512; heads $\mu, \log \sigma^2$ ($z$ per Tbl. 21) | FC: 512, 512 → reshape to $1 \times 28 \times 28$ |
| MLP | CIFAR-10 | FC: 1024, 1024 | FC: 1024, 1024 → $3 \times 32 \times 32$ |
| Conv | MNIST / FashionMNIST | 3×3/s2: 32, 64, 128 | Deconv s2: 128, 64, 32 → $C$ |
| Conv | CIFAR-10 | 4×4/s2: 64, 128, 256, 256 | Deconv s2: 256, 256, 128, 64 → $C$ |
| ResConv | MNIST / FashionMNIST | ResDown s2: 32, 64, 128 | ResUp mirror; final 3×3 to $C$ |
| ResConv | CIFAR-10 | ResDown s2: 64, 128, 256, 256 | ResUp mirror; final 3×3 to $C$ |

**WGAN-GP (projection critic).** Tables 23–25 summarize preprocessing deviations, training hyperparameters, and architectures.

Table 23: **WGAN-GP** data preprocessing deviations relative to Table 34.

| Dataset | Train input | Export size | Aug | Notes |
|---|---|---|---|---|
| MNIST | $1 \times 32 \times 32$ (28→32) | 28×28 | none | Center-crop 32→28 on export; scale $[-1, 1]$. |
| FashionMNIST | $1 \times 32 \times 32$ (28→32) | 28×28 | none | Center-crop 32→28 on export; scale $[-1, 1]$. |
| CIFAR-10 | $3 \times 32 \times 32$ | 32×32 | none | No cropping; scale $[-1, 1]$. |

Table 24: **WGAN-GP** training hyperparameters (shared across datasets).

| Ep | Batch | LR | $z$ | Cond | $G$ base | $D$ base | $D$ feat | $n_{\text{critic}}$ | $\lambda_{\text{GP}}$ | Adam $(\beta_1, \beta_2)$ |
|---|---|---|---|---|---|---|---|---|---|---|
| 100 | 64 | $1 \times 10^{-4}$ | 128 | 128 | 256 | 64 | 256 | 5 | 10.0 | (0.0, 0.9) |

Table 25: **WGAN-GP** model: DCGAN-style conditional generator and projection critic (Miyato and Koyama, 2018).

| | |
|---|---|
| Generator | $z \in \mathbb{R}^{128}$, label emb. $\in \mathbb{R}^{128}$; concat → FC to $4 \times 4 \times 256$; deconvs: 256→128→64→32 (stride 2, BN+ReLU); 3×3 conv → $C$; tanh output in $[-1, 1]$. |
| Critic (proj) | Convs: $C$→64→128→256 (stride 2, LeakyReLU 0.2) to 4×4; global sum pool → $\mathbb{R}^{256}$; score $= w^\top h + \langle h, E[y] \rangle$ with class embedding $E \in \mathbb{R}^{K \times 256}$. |
| Objective | WGAN-GP with projection discriminator; gradient penalty $\lambda_{\text{GP}} = 10$; $n_{\text{critic}} = 5$. |
| Exports | Per-class sampler: generates $[-1, 1]$ then maps to $[0, 1]$; MNIST/FashionMNIST center-cropped 32→28 on save; CIFAR-10 kept at 32×32. |

**DCGAN and StyleGAN2-lite.** Preprocessing and training settings are summarized in Tables 26–28.

Table 26: Data preprocessing deviations for **DCGAN** (Radford et al., 2016) and **StyleGAN2-lite** (Karras et al., 2020) relative to Table 34.

| Model | Dataset | Train input | Export size | Notes |
|---|---|---|---|---|
| DCGAN | MNIST / FashionMNIST | $1{\times}28{\times}28$ | $28{\times}28$ | No aug; scale $[-1, 1]$. |
| DCGAN | CIFAR-10 | $3{\times}32{\times}32$ | $32{\times}32$ | No aug; scale $[-1, 1]$. |
| StyleGAN2-lite | MNIST / FashionMNIST | $1{\times}32{\times}32$ ($28{\to}32$) | $32{\times}32$ | SG2-lite fixed to $32^2$; no export crop. |
| StyleGAN2-lite | CIFAR-10 | $3{\times}32{\times}32$ | $32{\times}32$ | SG2-lite fixed to $32^2$. |

Table 27: Training hyperparameters for **DCGAN** and **StyleGAN2-lite**.

| Model | Dataset | Epochs | Batch | $z$ | Cond. dim | $\text{LR}_G$ | $\text{LR}_D$ | Adam $(\beta_1, \beta_2)$ | Reg. | EMA |
|---|---|---|---|---|---|---|---|---|---|---|
| DCGAN | MNIST | 100 | 128 | 128 | 64 | $2{\times}10^{-4}$ | $2{\times}10^{-4}$ | (0.5, 0.999) | BCE (real=0.9) | 0.999 |
| | FashionMNIST | 120 | 128 | 128 | 64 | $2{\times}10^{-4}$ | $2{\times}10^{-4}$ | (0.5, 0.999) | BCE (real=0.9) | 0.999 |
| | CIFAR-10 | 200 | 128 | 128 | 64 | $2{\times}10^{-4}$ | $2{\times}10^{-4}$ | (0.5, 0.999) | BCE (real=0.9) | 0.999 |
| StyleGAN2-lite | MNIST | 100 | 128 | 128 | 64 | $2{\times}10^{-4}$ | $2{\times}10^{-4}$ | (0.0, 0.99) | R1 ($\gamma{=}10$) | 0.999 |
| | FashionMNIST | 120 | 128 | 128 | 64 | $2{\times}10^{-4}$ | $2{\times}10^{-4}$ | (0.0, 0.99) | R1 ($\gamma{=}10$) | 0.999 |
| | CIFAR-10 | 200 | 128 | 128 | 64 | $2{\times}10^{-4}$ | $2{\times}10^{-4}$ | (0.0, 0.99) | R1 ($\gamma{=}10$) | 0.999 |

Table 28: **DCGAN** and **StyleGAN2-lite** architectures. $C{=}1$ (MNIST/Fashion-MNIST), $C{=}3$ (CIFAR-10).

| **DCGAN (cond.)** | **G:** $[z\,|\,\text{emb}(y)] \to$ FC $\to$ seed ($7 \times 7$ if $28 \times 28$, else $8 \times 8$), width $g_{\text{ch}}{=}128$; two ConvT $4 \times 4$, $s{=}2$ (BN+ReLU) to $g_{\text{ch}}/4$; final $3 \times 3$ conv $\to C$; tanh. **D:** concat one-hot (10 ch) on input; Conv $4 \times 4$, $s{=}2$: $C{+}10 \to d_{\text{ch}}/2 \to d_{\text{ch}}$ (BN+LReLU 0.2); final conv $k{=}7/8 \to$ logit. |
|---|---|
| **StyleGAN2-lite (cond., $32 \times 32$)** | **G:** MLP on $[z\,|\,\text{emb}(y)] \to w$ (256-d, 4$\times$ LReLU); learned $4 \times 4 \times g_{\text{ch}}$ constant; StyledConv (mod+demod, per-layer noise) with upsampling $4 \to 8 \to 16 \to 32$; skip-sum `ToRGB` at 4/8/16/32; tanh. **D:** `FromRGB` $\to$ residual downs $32 \to 16 \to 8 \to 4$ (avg-pool); final convs; projection: scalar $+\langle\text{proj}(h), \text{emb}(y)\rangle$ (256-d). |

**Conditional diffusion models (cDDPM).** We train both UNet- and Transformer2D-based cDDPMs with classifier-free guidance and evaluate using fast samplers at test time. Training and model configurations appear in Tables 29–33.

Table 29: Optimization and diffusion schedule for **cDDPM** (all datasets). Data/splits follow Table 34.

| Datasets | Epochs | Batch | Optim | LR | Timesteps | Pred. type | EMA | Clip | Seed |
|---|---|---|---|---|---|---|---|---|---|
| MNIST / FashionMNIST / CIFAR-10 | 50 | 128 | AdamW | $1{\times}10^{-4}$ | 1000 | v-pred | 0.999 | 1.0 | 42 |

Table 30: UNet config, conditioning, and sampling for **cDDPM** (only $C/H/W$ vary by dataset).

| UNet blocks | Stage ch. | Heads | Cond dim | $p_{\text{uncond}}$ | Sampler | Steps | CFG | Preview |
|---|---|---|---|---|---|---|---|---|
| $3{\downarrow}/3{\uparrow}$, 2 layers (cross-attn) | (128, 256, 256) | 4 | 128 | 0.10 | DPM-Solver++ | 50 | 2.0 | 10/class |

Table 31: Training hyperparameters for **Transformer2D cDDPM**. Optimizer: AdamW ($\beta_1{=}0.9, \beta_2{=}0.999$); loss: MSE on $v$ (v-prediction).

| Dataset | Epochs | Batch | LR | Weight decay | Timesteps | Pred. type | EMA decay | Sample steps | Seed |
|---|---|---|---|---|---|---|---|---|---|
| MNIST | 150 | 256 | $1{\times}10^{-4}$ | 0.0 | 1000 | v-pred | 0.9999 | 16 | 42 |
| FashionMNIST | 160 | 256 | $1{\times}10^{-4}$ | 0.0 | 1000 | v-pred | 0.9999 | 20 | 42 |
| CIFAR-10 | 400 | 256 | $1{\times}10^{-4}$ | 0.0 | 1000 | v-pred | 0.9999 | 20 | 42 |

Table 32: Model and conditioning for **Transformer2D cDDPM**. Scheduler (train): DDPM with linear $\beta$ $(10^{-4} \to 2 \times 10^{-2})$; sampler: DPM-Solver++ (CFG).

| Dataset | Patch | Layers | Heads | Head dim | Cond dim | $p_{\mathrm{uncond}}$ | CFG scale | In ch | Size |
|---|---|---|---|---|---|---|---|---|---|
| MNIST | 2 | 8 | 4 | 64 | 128 | 0.20 | 2.0 | 1 | 28 |
| FashionMNIST | 2 | 10 | 4 | 64 | 128 | 0.20 | 2.0 | 1 | 28 |
| CIFAR-10 | 4 | 12 | 6 | 64 | 128 | 0.20 | 2.0 | 3 | 32 |

Table 33: Model and sampling configuration for **UNet cDDPM**. Training scheduler: DDPM (squared-cosine $\beta$, v-pred); sampler: Euler Ancestral with CFG.

| Dataset | Blocks (ch.) | Layers/block | In/Out ch. | Size | Class slots | Cond. type | Pred. type | Sampler |
|---|---|---|---|---|---|---|---|---|
| MNIST / FashionMNIST | (128, 256, 256) | 2 | $1 \to 1$ | 28 | $K{+}1$ (0=null) | class_embed (timestep) | v-pred | EulerA |
| CIFAR-10 | (128, 256, 256) or (128, 256, 512)[†] | 2 | $3 \to 3$ | 32 | $K{+}1$ (0=null) | class_embed (timestep) | v-pred | EulerA |

† `--wide` switches CIFAR-10 to $(128, 256, 512)$; `--channels` accepts a custom comma list (e.g., `128,256,256`).

Classifier-free guidance: during training labels drop to the null class with prob. $p_{\mathrm{uncond}}$; at sampling, batches are doubled with uncond/cond labels (0, $c{+}1$) and combined as $u + s(c-u)$.

# 18 Example Generated Molecules by Each Generator

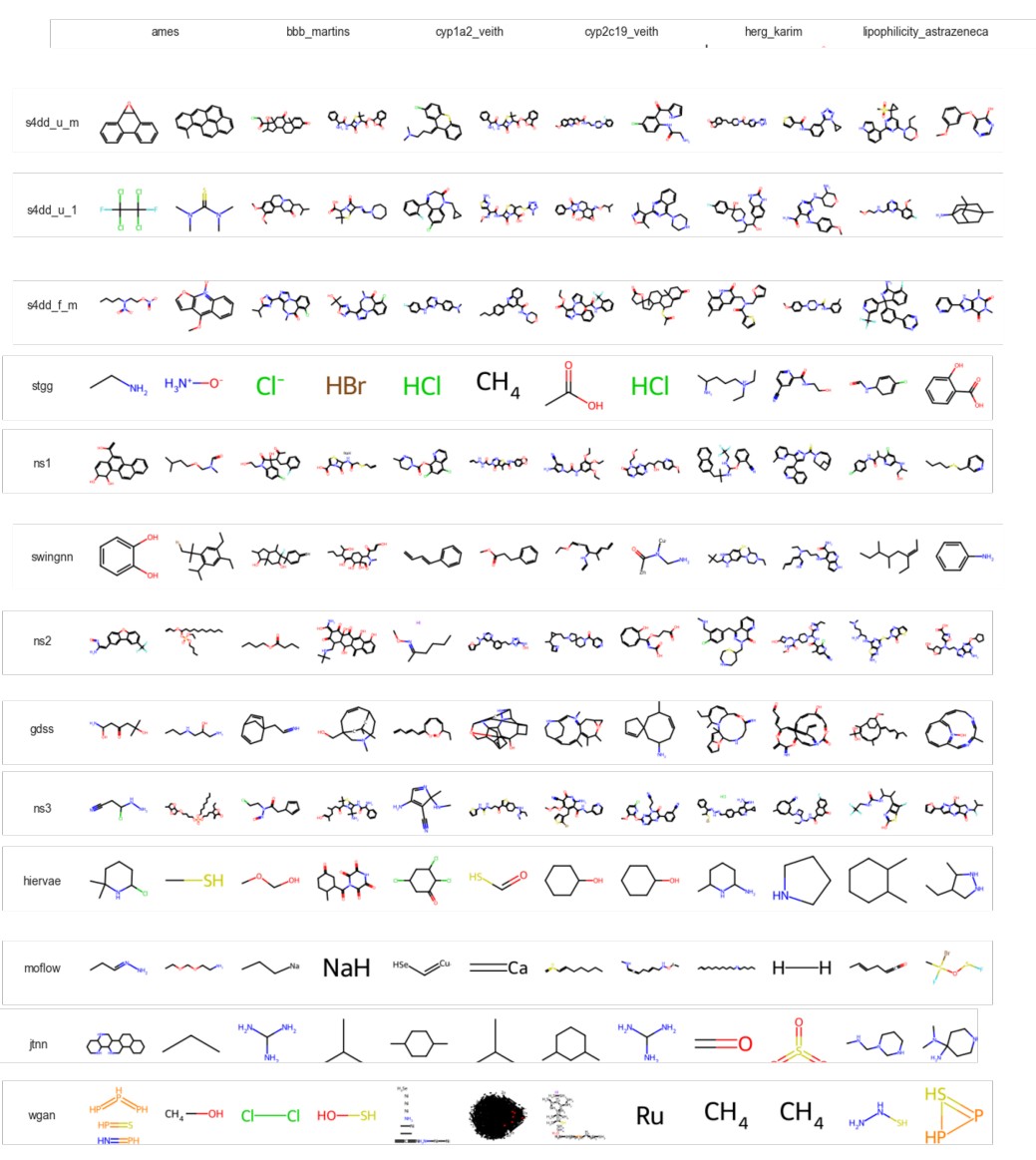

Figure 4: Examples of molecules generated by different models across the six datasets. Each row is a generator; each dataset contributes a positive and a negative sample.

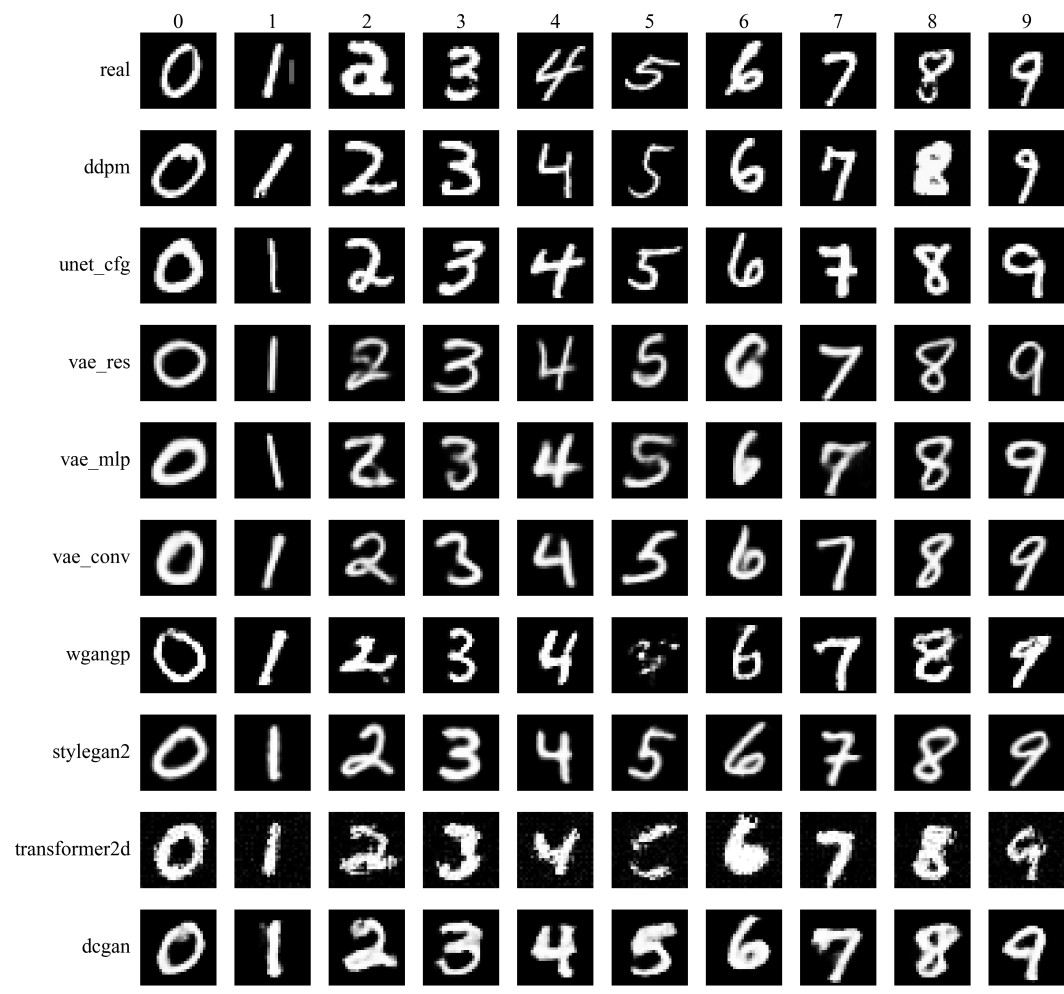

Figure 5: Samples generated by each model on **MNIST**. Rows: generators; columns: class-conditional samples (uncurated).

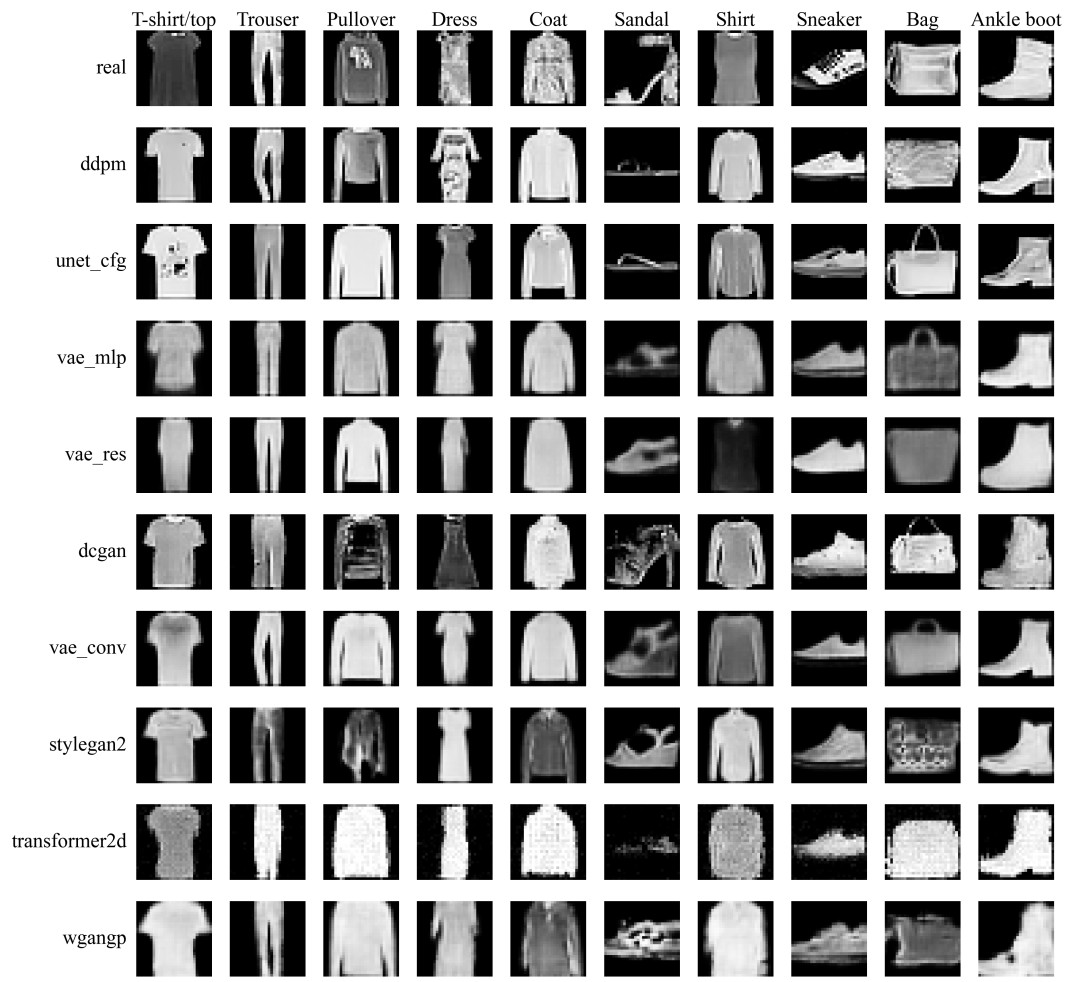

Figure 6: Samples generated by each model on **Fashion–MNIST**. Rows: generators; columns: class-conditional samples (uncurated).

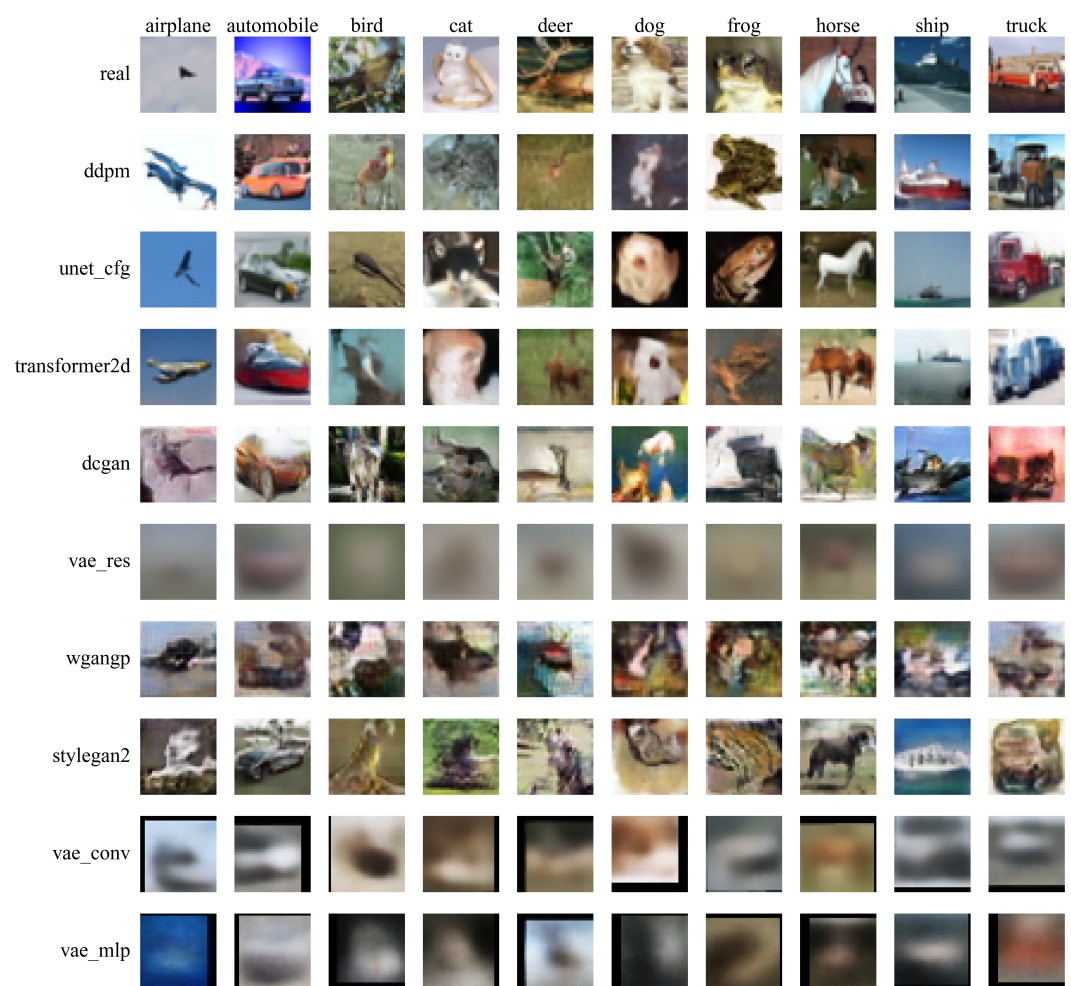

Figure 7: Samples generated by each model on **CIFAR–10**. Rows: generators; columns: per-class conditional samples (uncurated).

## 19 PRECISION, RECALL, DENSITY, AND COVERAGE RANK OF REAL GENERATORS

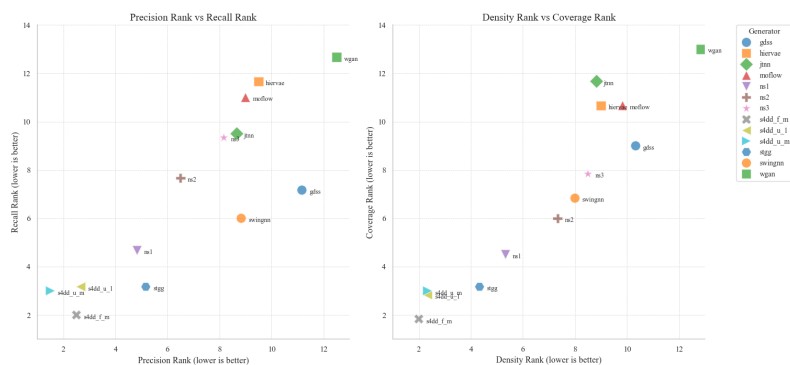

Figure 8: Rank comparisons of real graph generators: Precision vs. Recall (left) and Density vs. Coverage (right). Lower rank is better.

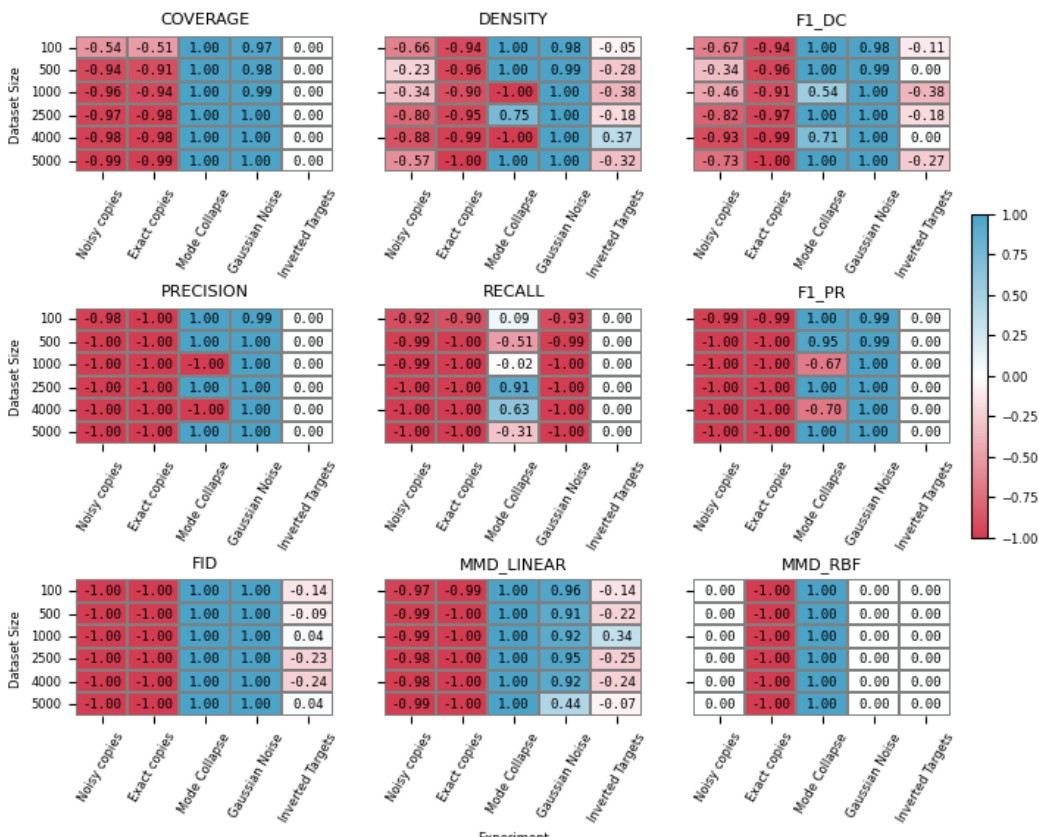

Figure 9: Effect of perturbations on Coverage, Density, F1-DC, Precision, Recall, F1-PR, FID, MMD_Linear, MMD_RBF across dataset sizes.

Despite broad use, many alternative metrics show limited robustness to perturbations; e.g., MMD_RBF is nearly unchanged under noisy copies.

# 21 Real-World Dataset Generation Experiments in Detail

## 21.1 Average Rank per Dataset for Each Vectorizer

### 21.1.1 AMES

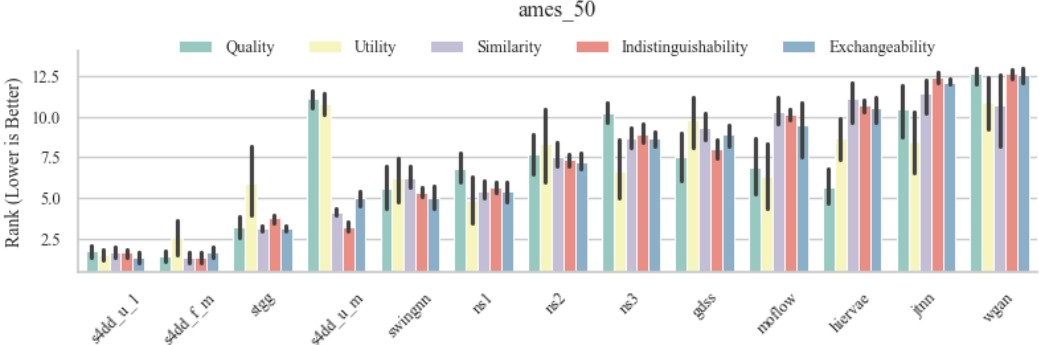

Figure 10: Average generator rank on **AMES** across vectorizers (lower is better).

### 21.1.2 BBB Martins

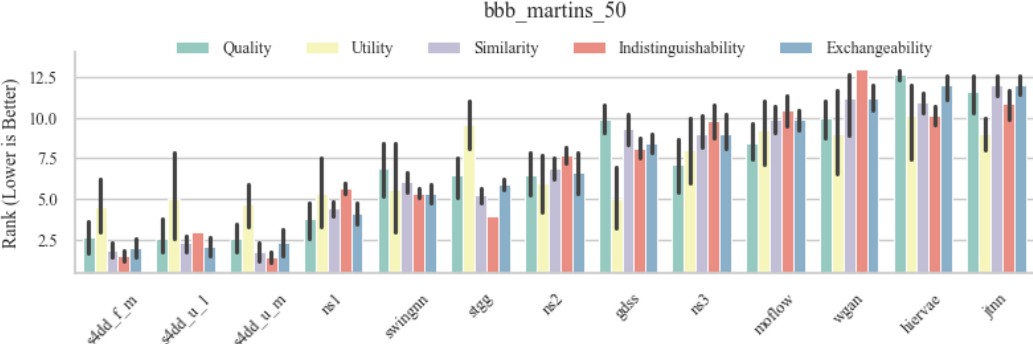

Figure 11: Average generator rank on **BBB Martins** across vectorizers (lower is better).

### 21.1.3 CYP1A2

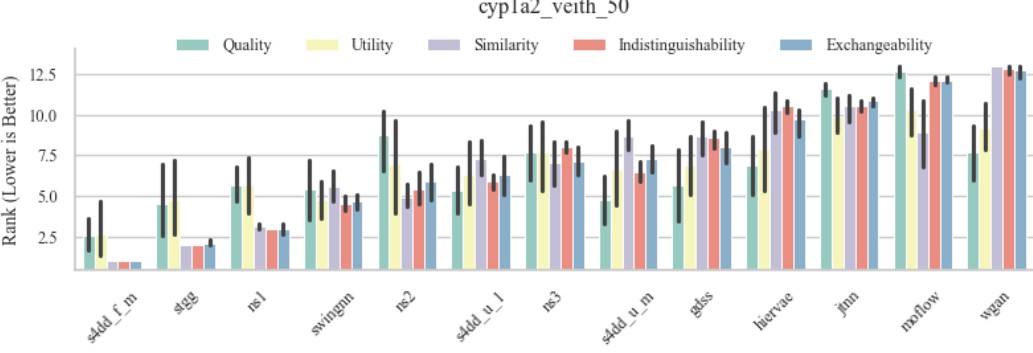

Figure 12: Average generator rank on **CYP1A2** across vectorizers (lower is better).

### 21.1.4  CYP2C19

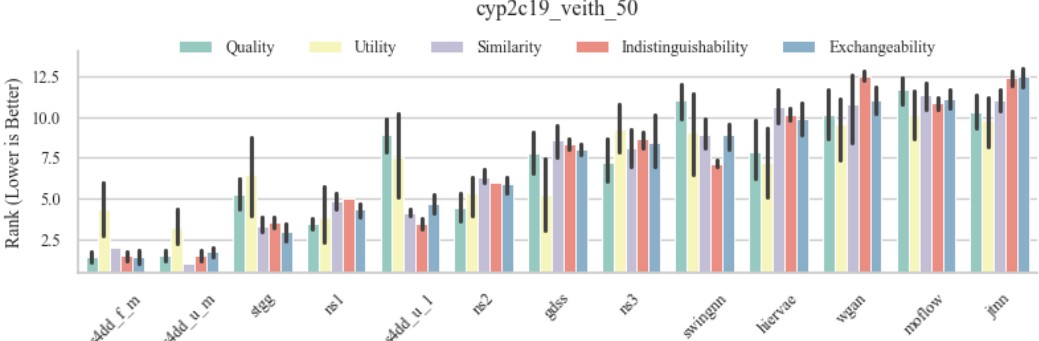

Figure 13: Average generator rank on **CYP2C19** across vectorizers (lower is better).

### 21.1.5  HERG KARIM

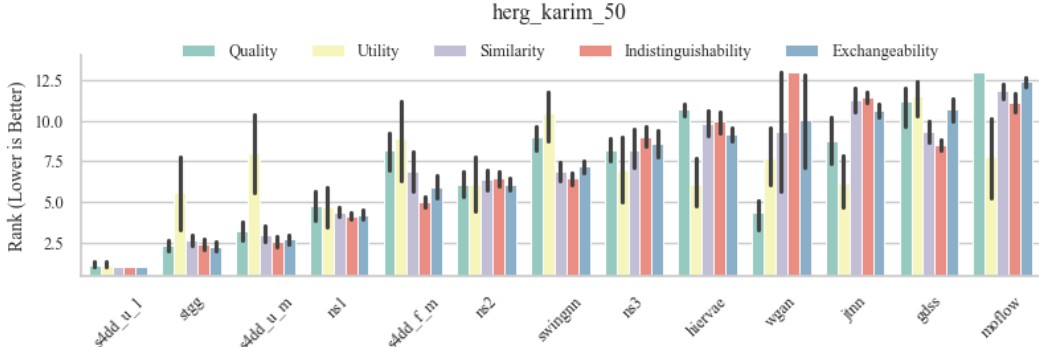

Figure 14: Average generator rank on **hERG Karim** across vectorizers (lower is better).

### 21.1.6  LIPOPHILICITY ASTRAZENECA

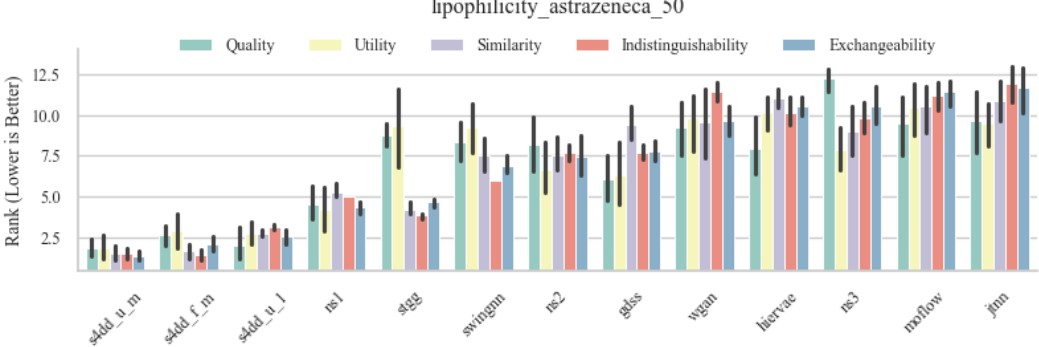

Figure 15: Average generator rank on **Lipophilicity AstraZeneca** across vectorizers (lower is better).

## 22 Image-Domain Experiments: Datasets, Models, and Full Results

### 22.1 Datasets, Preprocessing, and Splits

Table 34: Shared preprocessing and canonical splits used for all models and experiments.

| Dataset | Shape $(C \times H \times W)$ | Pixel scale | Default train aug. | Split policy |
|---------|------------------------------|-------------|--------------------|--------------|
| MNIST | $1 \times 28 \times 28$ | $[-1, 1]$ from $[0, 1]$ | none | stratified 50/50 `train1`/`train2` (seed 42); test untouched |
| FashionMNIST | $1 \times 28 \times 28$ | $[-1, 1]$ | none | same |
| CIFAR-10 | $3 \times 32 \times 32$ | $[-1, 1]$ | none[†] | same |

[†] Some models enable flip/crop at training time; exports match dataset size.

### 22.2 Image Data Full Metric Results Tables

Table 35: Metric values (left) and ranks (right) on **CIFAR-10**. Lower rank is better.

| Generator | Q val | Q rank | U val | U rank | S val | S rank | I val | I rank | E val | E rank |
|-----------|-------|--------|-------|--------|-------|--------|-------|--------|-------|--------|
| DCGAN | 0.670 | 9.0 | 0.000 | 6.5 | 0.450 | 4.0 | 0.450 | 1.0 | 0.150 | 4.0 |
| DDPM | 0.940 | 2.5 | 0.250 | 3.0 | 0.550 | 1.0 | 0.430 | 2.0 | 0.290 | 1.0 |
| StyleGAN2 | 0.880 | 4.0 | 0.000 | 6.5 | 0.190 | 9.0 | 0.080 | 6.0 | 0.060 | 7.0 |
| Transformer2D | 0.950 | 1.0 | 0.290 | 1.0 | 0.510 | 3.0 | 0.400 | 4.0 | 0.280 | 3.0 |
| UNet_CFG | 0.940 | 2.5 | 0.250 | 2.0 | 0.530 | 2.0 | 0.410 | 3.0 | 0.280 | 2.0 |
| VAE_Conv | 0.760 | 6.0 | 0.000 | 6.5 | 0.230 | 7.0 | 0.010 | 7.5 | 0.050 | 8.0 |
| VAE_MLP | 0.740 | 7.0 | 0.000 | 6.5 | 0.210 | 8.0 | 0.010 | 7.5 | 0.040 | 9.0 |
| VAE_Res | 0.770 | 5.0 | 0.000 | 6.5 | 0.400 | 5.0 | 0.000 | 9.0 | 0.080 | 5.5 |
| WGAN-GP | 0.720 | 8.0 | 0.000 | 6.5 | 0.280 | 6.0 | 0.140 | 5.0 | 0.080 | 5.5 |

Table 36: Metric values (left) and ranks (right) on **Fashion-MNIST**. Lower rank is better.

| Generator | Q val | Q rank | U val | U rank | S val | S rank | I val | I rank | E val | E rank |
|-----------|-------|--------|-------|--------|-------|--------|-------|--------|-------|--------|
| DCGAN | 0.930 | 5.5 | 0.020 | 1.0 | 0.300 | 6.0 | 0.100 | 3.0 | 0.090 | 5.5 |
| DDPM | 0.940 | 2.5 | 0.000 | 5.5 | 0.500 | 1.5 | 0.170 | 1.0 | 0.160 | 1.0 |
| StyleGAN2 | 0.940 | 2.5 | 0.000 | 5.5 | 0.220 | 7.0 | 0.000 | 7.0 | 0.050 | 7.0 |
| Transformer2D | 0.810 | 8.0 | 0.000 | 5.5 | 0.060 | 8.0 | 0.000 | 7.0 | 0.010 | 8.5 |
| UNet_CFG | 0.940 | 2.5 | 0.000 | 5.5 | 0.500 | 1.5 | 0.140 | 2.0 | 0.150 | 2.0 |
| VAE_Conv | 0.920 | 7.0 | 0.000 | 5.5 | 0.400 | 5.0 | 0.000 | 7.0 | 0.090 | 5.5 |
| VAE_MLP | 0.940 | 2.5 | 0.000 | 5.5 | 0.430 | 3.5 | 0.000 | 7.0 | 0.100 | 3.5 |
| VAE_Res | 0.930 | 5.5 | 0.000 | 5.5 | 0.430 | 3.5 | 0.000 | 7.0 | 0.100 | 3.5 |
| WGAN-GP | 0.790 | 9.0 | 0.000 | 5.5 | 0.050 | 9.0 | 0.010 | 4.0 | 0.010 | 8.5 |

Table 37: Metric values (left) and ranks (right) on **MNIST**. Lower rank is better.

| Generator | Q val | Q rank | U val | U rank | S val | S rank | I val | I rank | E val | E rank |
|-----------|-------|--------|-------|--------|-------|--------|-------|--------|-------|--------|
| dcgan | 0.560 | 9.0 | 0.000 | 7.0 | 0.010 | 9.0 | 0.000 | 6.0 | 0.000 | 9.0 |
| ddpm | 0.970 | 3.0 | 0.000 | 7.0 | 0.570 | 2.0 | 0.130 | 1.5 | 0.170 | 1.5 |
| stylegan2 | 0.920 | 7.0 | 0.090 | 1.0 | 0.090 | 7.0 | 0.060 | 6.0 | 0.020 | 7.0 |
| transformer2d | 0.820 | 8.0 | 0.000 | 7.0 | 0.070 | 8.0 | 0.000 | 6.0 | 0.020 | 8.0 |
| unet_cfg | 0.970 | 3.0 | 0.000 | 7.0 | 0.580 | 1.0 | 0.130 | 1.5 | 0.170 | 1.5 |
| vae_conv | 0.960 | 6.0 | 0.000 | 7.0 | 0.480 | 5.0 | 0.000 | 6.0 | 0.110 | 5.0 |
| vae_mlp | 0.970 | 3.0 | 0.000 | 7.0 | 0.490 | 4.0 | 0.000 | 6.0 | 0.120 | 4.0 |
| vae_res | 0.970 | 3.0 | 0.000 | 4.0 | 0.520 | 3.0 | 0.000 | 6.0 | 0.130 | 3.0 |
| wgangp | 0.970 | 3.0 | 0.090 | 2.0 | 0.400 | 6.0 | 0.000 | 6.0 | 0.110 | 6.0 |

Table 38: CIFAR-10 metrics with per-metric ranks ($r=1$ best). $n_{\text{ref}}$=25,000, $n_{\text{gen}}$=25,000.

| Model | FID ↓ | r | KID ↓ | r | P | r | R | r | F1_pr | r | IS | r |
|---|---|---|---|---|---|---|---|---|---|---|---|---|
| DCGAN | 21.62 | 5 | 0.0171 | 5 | 0.165 | 6 | 0.078 | 3 | 0.106 | 3 | 2.041 | 1 |
| DDPM | 6.67 | 1 | 0.0030 | 1 | 0.224 | 3 | 0.122 | 1 | 0.158 | 1 | 2.017 | 2 |
| StyleGAN2 | 19.26 | 3 | 0.0129 | 3 | 0.220 | 4 | 0.002 | 6 | 0.005 | 6 | 1.958 | 3 |
| UNet_CFG | 9.62 | 2 | 0.0069 | 2 | 0.288 | 1 | 0.093 | 2 | 0.141 | 2 | 1.944 | 4 |
| Transformer2D | 20.13 | 4 | 0.0150 | 4 | 0.286 | 2 | 0.065 | 4 | 0.106 | 3 | 1.763 | 5 |
| VAE_Conv | 182.67 | 7 | 0.1905 | 7 | 0.110 | 7 | 0.000 | 7 | 0.000 | 7 | 1.293 | 7 |
| VAE_MLP | 206.49 | 8 | 0.2182 | 8 | 0.094 | 8 | 0.000 | 7 | 0.000 | 7 | 1.280 | 8 |
| VAE_Res | 374.51 | 9 | 0.4439 | 9 | 0.002 | 9 | 0.000 | 7 | 0.000 | 7 | 1.006 | 9 |
| WGAN-GP | 42.62 | 6 | 0.0342 | 6 | 0.207 | 5 | 0.017 | 5 | 0.032 | 5 | 1.652 | 6 |

Table 39: Fashion-MNIST metrics with per-metric ranks ($r=1$ best). $n_{\text{ref}}$=30,000, $n_{\text{gen}}$=30,000.

| Model | FID ↓ | r | KID ↓ | r | P | r | R | r | F1_pr | r | IS | r |
|---|---|---|---|---|---|---|---|---|---|---|---|---|
| DCGAN | 0.19 | 1 | 0.0106 | 4 | 0.215 | 7 | 0.174 | 3 | 0.192 | 3 | 2.364 | 1 |
| DDPM | 0.71 | 2 | 0.0018 | 1 | 0.369 | 2 | 0.246 | 1 | 0.295 | 1 | 1.994 | 3 |
| StyleGAN2 | 1.45 | 4 | 0.0077 | 3 | 0.318 | 3 | 0.001 | 8 | 0.002 | 8 | 1.869 | 5 |
| UNet_CFG | 1.10 | 3 | 0.0042 | 2 | 0.463 | 1 | 0.188 | 2 | 0.268 | 2 | 1.905 | 4 |
| Transformer2D | 5.45 | 9 | 0.0587 | 9 | 0.090 | 8 | 0.001 | 8 | 0.002 | 8 | 1.570 | 9 |
| VAE_Conv | 3.82 | 6 | 0.0231 | 6 | 0.230 | 6 | 0.009 | 4 | 0.017 | 4 | 1.692 | 6 |
| VAE_MLP | 4.59 | 8 | 0.0216 | 5 | 0.252 | 5 | 0.007 | 5 | 0.014 | 5 | 1.677 | 7 |
| VAE_Res | 3.99 | 7 | 0.0257 | 7 | 0.257 | 4 | 0.006 | 6 | 0.011 | 6 | 1.673 | 8 |
| WGAN-GP | 2.19 | 5 | 0.0423 | 8 | 0.057 | 9 | 0.005 | 7 | 0.009 | 7 | 2.013 | 2 |

Table 40: MNIST metrics with per-metric ranks ($r=1$ best). $n_{\text{ref}}$=30,000, $n_{\text{gen}}$=30,000.

| Model | FID ↓ | r | KID ↓ | r | P | r | R | r | F1_pr | r | IS | r |
|---|---|---|---|---|---|---|---|---|---|---|---|---|
| DCGAN | 3.89 | 9 | 0.0321 | 8 | 0.307 | 3 | 0.003 | 7 | 0.007 | 7 | 1.276 | 9 |
| DDPM | 0.25 | 4 | 0.0016 | 1 | 0.488 | 2 | 0.319 | 1 | 0.386 | 1 | 1.634 | 7 |
| StyleGAN2 | 2.27 | 8 | 0.0144 | 6 | 0.291 | 4 | 0.000 | 9 | 0.000 | 9 | 1.478 | 8 |
| UNet_CFG | 0.29 | 5 | 0.0017 | 2 | 0.531 | 1 | 0.299 | 2 | 0.383 | 2 | 1.636 | 6 |
| Transformer2D | 0.83 | 7 | 0.1345 | 9 | 0.027 | 9 | 0.003 | 7 | 0.005 | 8 | 1.647 | 5 |
| VAE_Conv | 0.24 | 3 | 0.0114 | 3 | 0.223 | 5 | 0.072 | 5 | 0.109 | 5 | 1.784 | 3 |
| VAE_MLP | 0.23 | 2 | 0.0125 | 4 | 0.206 | 8 | 0.082 | 4 | 0.117 | 4 | 1.816 | 2 |
| VAE_Res | 0.40 | 6 | 0.0141 | 5 | 0.212 | 6 | 0.052 | 6 | 0.084 | 6 | 1.716 | 4 |
| WGAN-GP | 0.14 | 1 | 0.0163 | 7 | 0.210 | 7 | 0.207 | 3 | 0.209 | 3 | 1.865 | 1 |

Table 41: Kernel-entropy metrics on CIFAR-10 with per-metric ranks ($r=1$ best). Higher is better for RKE_gen, FKEA-VENDI_gen, FKEA-RKE2_gen; lower is better for RRKE.

| Model | $\sigma$ | RKE_gen | r | $\hat{m}$ | RRKE | r | FKEA-VENDI_gen | r | FKEA-RKE2_gen | r |
|---|---|---|---|---|---|---|---|---|---|---|
| DCGAN | 15.18 | 0.852 | 5 | 2 | 0.179 | 5 | 13.1 | 3 | 2.3 | 3 |
| DDPM | 15.72 | 0.942 | 1 | 3 | 0.120 | 1 | 15.7 | 1 | 2.6 | 1 |
| StyleGAN2 | 15.21 | 0.858 | 3 | 2 | 0.167 | 4 | 12.6 | 4 | 2.3 | 3 |
| UNet_CFG | 15.54 | 0.901 | 2 | 2 | 0.129 | 2 | 14.0 | 2 | 2.4 | 2 |
| Transformer2D | 15.28 | 0.855 | 4 | 2 | 0.165 | 3 | 12.4 | 5 | 2.4 | 2 |
| VAE_Conv | 15.34 | 0.335 | 7 | 1 | 0.762 | 7 | 2.8 | 7 | 1.4 | 5 |
| VAE_MLP | 15.54 | 0.305 | 8 | 1 | 0.829 | 8 | 2.5 | 8 | 1.4 | 5 |
| VAE_Res | 17.04 | 0.014 | 9 | 1 | 1.228 | 9 | 1.1 | 9 | 1.0 | 6 |
| WGAN-GP | 14.82 | 0.745 | 6 | 2 | 0.249 | 6 | 9.5 | 6 | 2.1 | 4 |

Table 42: Kernel-entropy metrics on Fashion-MNIST with per-metric ranks ($r$=1 best).

| Model | $\sigma$ | RKE_gen | r | $\hat{m}$ | RRKE | r | FKEA-VENDI_gen | r | FKEA-RKE2_gen | r |
|---|---|---|---|---|---|---|---|---|---|---|
| DCGAN | 16.63 | 0.953 | 1 | 3 | 0.120 | 3 | 14.7 | 1 | 2.6 | 1 |
| DDPM | 16.56 | 0.945 | 2 | 3 | 0.083 | 1 | 13.1 | 2 | 2.6 | 1 |
| StyleGAN2 | 16.39 | 0.905 | 4 | 2 | 0.133 | 4 | 11.1 | 4 | 2.5 | 2 |
| UNet_CFG | 16.46 | 0.923 | 3 | 3 | 0.092 | 2 | 11.9 | 3 | 2.5 | 2 |
| Transformer2D | 16.62 | 0.848 | 7 | 2 | 0.357 | 9 | 8.7 | 9 | 2.3 | 4 |
| VAE_Conv | 16.19 | 0.852 | 6 | 2 | 0.205 | 6 | 9.3 | 6 | 2.3 | 4 |
| VAE_MLP | 16.26 | 0.852 | 6 | 2 | 0.199 | 5 | 9.0 | 8 | 2.3 | 4 |
| VAE_Res | 16.12 | 0.837 | 8 | 2 | 0.211 | 7 | 9.1 | 7 | 2.3 | 4 |
| WGAN-GP | 16.54 | 0.879 | 5 | 2 | 0.263 | 8 | 10.6 | 5 | 2.4 | 3 |

Table 43: Kernel-entropy metrics on MNIST with per-metric ranks ($r$=1 best).

| Model | $\sigma$ | RKE_gen | r | $\hat{m}$ | RRKE | r | FKEA-VENDI_gen | r | FKEA-RKE2_gen | r |
|---|---|---|---|---|---|---|---|---|---|---|
| DCGAN | 13.58 | 0.869 | 8 | 2 | 0.331 | 8 | 8.1 | 8 | 2.3 | 5 |
| DDPM | 13.48 | 0.971 | 3 | 3 | 0.070 | 1 | 13.5 | 2 | 2.7 | 1 |
| StyleGAN2 | 13.21 | 0.882 | 7 | 2 | 0.235 | 7 | 8.7 | 7 | 2.4 | 4 |
| UNet_CFG | 13.40 | 0.972 | 2 | 3 | 0.071 | 2 | 13.0 | 3 | 2.6 | 2 |
| Transformer2D | 15.28 | 0.850 | 9 | 2 | 0.540 | 9 | 10.7 | 6 | 2.3 | 5 |
| VAE_Conv | 13.46 | 0.937 | 5 | 3 | 0.146 | 4 | 12.5 | 4 | 2.5 | 3 |
| VAE_MLP | 13.67 | 0.953 | 4 | 3 | 0.150 | 5 | 13.0 | 3 | 2.6 | 2 |
| VAE_Res | 13.38 | 0.921 | 6 | 3 | 0.161 | 6 | 11.9 | 5 | 2.5 | 3 |
| WGAN-GP | 13.93 | 0.994 | 1 | 3 | 0.145 | 3 | 15.0 | 1 | 2.7 | 1 |

**Relationship between RankGen and conventional metrics.** Tables 35–37(RankGen metrics) and Tables 38–43(conventional metrics) show that the two families of metrics agree at a coarse level but diverge in precisely the situations where standard metrics are known to be unreliable. Across CIFAR–10, Fashion–MNIST, and MNIST, diffusion models (DDPM, UNet–CFG) rank near the top under both RankGen and baseline metrics such as FID (Heusel et al., 2017), KID (Binkowski et al., 2018), precision/recall for distributions (Sajjadi et al., 2018; Kynkäänniemi et al., 2019) and Inception Score (Salimans et al., 2016), as well as VENDI/RKE/FKEA kernel–entropy metrics (Friedman and Dieng, 2023; Pasarkar and Dieng, 2024; Jalali et al., 2024; Luo et al., 2024). However, the generators that the conventional metrics disagree on are exactly the ones where RankGen provides additional diagnostic resolution. For example, StyleGAN2 achieves competitive FID/KID and very high precision but extremely low recall, whereas RankGen penalizes it strongly through low Utility and low Similarity, revealing its concentrated, mode-dropping behaviour. Conversely, WGAN–GP obtains middling FID/KID but is classified by RankGen as having almost zero Utility, correctly indicating that its samples add little task-relevant information despite looking plausible in feature space. VAEs rank moderately under FID/KID but RankGen identifies their weak task fidelity (low Quality) and weak local mixing (low Indistinguishability), especially on CIFAR–10. Overall, baseline metrics capture global similarity, while RankGen captures *task fidelity, augmentation value, local mixing, and copying*, explaining the ranking discrepancies and providing complementary diagnostic information.

## 22.3 Reproducibility Notes

All experiments use seed 42 unless otherwise stated. Evaluation uses equal $n_{\mathrm{ref}}$ and $n_{\mathrm{gen}}$ per dataset (see captions).

## 23 Practical Capacity Choices for the PAC Bounds

In our experiments we do not attempt to estimate data-dependent Rademacher complexities or exact VC dimensions for each classifier. Instead, following standard PAC practice, we

use simple, conservative analytic upper bounds that make the guarantees easy to implement and reproduce.

**Rademacher complexity for the Quality bound.** The Quality bound requires an upper bound on the empirical Rademacher complexity $\mathcal{R}_n$ of the task-classifier class. We use the following rule:

- If an upper bound $d$ on the VC dimension is supplied, we use the classical VC-based inequality

$$\mathcal{R}_n \leq \sqrt{\frac{2d \log(en/d)}{n}},$$

  which follows from the Sauer–Shelah lemma and standard symmetrization.
- If no VC bound is supplied, we fall back to the distribution-free surrogate

$$\mathcal{R}_n \approx \sqrt{\frac{2}{n}},$$

  which produces a valid (although loose) Rademacher upper bound.

For the synthetic experiments we optionally allow a small, model-specific constant $\mathcal{R}_n$ (e.g., 0.01–0.09) as a capacity surrogate for each classifier family. These values are chosen to be no larger than the trivial $\sqrt{2/n}$ upper bound, so the guarantees remain conservative. The effect of these choices is only to make the gate slightly tighter or looser; the qualitative patterns we report are stable across a wide range of such constants.

**VC dimension for discriminators in Utility and Indistinguishability.** The Utility and Indistinguishability bounds require a capacity term for the real-vs-generated discriminator class. Since exact VC dimensions for modern ensembles (e.g., shallow forests or small MLPs) are difficult to compute, we set a single conservative upper bound $d_{\mathrm{disc}} = 50$ for all discriminators. This value is passed to the bound functions as a hyperparameter; larger choices only weaken the bounds.

If $d_{\mathrm{disc}}$ is set to `None`, our implementation reverts to a Hoeffding-type deviation bound that depends only on the number of evaluation examples $m$,

$$\varepsilon(m, \delta) = \sqrt{\frac{\log(2/\delta)}{2m}},$$

which yields a completely distribution-free PAC guarantee at the cost of additional looseness.

**Role of these constants.** Across experiments, these capacity terms are used only to construct binary pass/fail gates for each generator–metric pair rather than to provide tight numerical certificates. Once a model passes the gate, ordering is based on robust quantile summaries (median/IQR) as described in Sec. 3.8. We verified that the qualitative conclusions in the paper (e.g., which models pass the gates and the relative ordering of surviving generators) are stable under reasonable variations of the capacity parameters.

