# OpenReview forum: "RankGen: A Statistically Robust Framework for Ranking Generative Models Using Classifier-Based Metrics"
_ICLR.cc/2026/Conference — Submitted to ICLR 2026_

### Official Review · Reviewer_kCSF · 2025-10-17

**Soundness:** 2
**Presentation:** 2
**Contribution:** 2
**Rating:** 4
**Confidence:** 4

**Summary:**

The paper introduces RankGen, a unified and statistically robust framework for evaluating and ranking generative models, designed to overcome the brittleness and shortcomings of standard, easy-to-game evaluation metrics. RankGen utilizes four PAC-style generalization-bound classifier-based metrics—Quality, Utility, Indistinguishability, and Similarity—to capture distinct failure modes like low fidelity, memorization, distribution shift, and mode collapse.

**Strengths:**

1. a new method different with the previous FID, IS, Precision&Recall
2. give the new definition for generative quality

**Weaknesses:**

1. The paper must clearly define the operational and conceptual distinction between Quality, Similarity, and Utility. . All are to discuss the diversity situation.
2. The paper uses "Quality" to assess distribution coverage (fit) rather than the standard meaning of sample fidelity (realism). This non-standard usage must be explicitly stated early on to avoid confusion with existing generative metrics (e.g., FID).
3. The discussion on the metric's diagnostic role, particularly concerning mode collapse (Sec 3.3, 3.6), needs significant expansion. This is a critical phenomenon that requires a deeper analysis, including differentiating between fidelity collapse and diversity collapse. The diagnostic role should be systematically extended to other failure modes (e.g., over-generalization, memorization).
4. similar to 3, the other Diagnostic role from sec 3.5 and 3.6  should also discuss more.
5. In the paper, the reliance on two specific training sets, ($train_1$ and $train_2$), severely limits applicability.
Unlabeled Data: The authors must propose a methodology to guarantee the distinguishability of  $train_1$ and $train_2$  when working with unlabeled datasets.
Conditional Models: The current Indistinguishability task may be inappropriate for labeled/conditional generative models. The authors must state this limitation or provide an adaptation for such models.
6. Figure 2 is not clear. and figure 3 should give the score for the different models (more clear to see the rank)

**Questions:**

see the weakness

---

> ### Author Response · Authors · 2025-11-24
>
> **Q1**
>
> We thank the reviewer for raising this point. We would like to clarify that in Section 3 of the paper, each of the four metrics already begins with a plain-English operational and conceptual description before any formulas are introduced. Specifically:
>
> * **Quality (Sec. 3.3)**:
>   We open with a natural-language explanation—*“The quality score measures how much predictive signal synthetic data preserves relative to real data.”*
>   This operationally defines Quality as a *task-relevant fidelity measure*.
>
> * **Utility (Sec. 3.4)**:
>   The section begins—*“The utility score measures how much useful and non-redundant information synthetic data provides when used to augment real training data.”*
>   This describes Utility as capturing *incremental, non-redundant diversity*.
>
> * **Indistinguishability (Sec. 3.5)**:
>   We introduce it with—*“The indistinguishability score asks how hard it is for a discriminator to tell real from generated samples.”*
>   This makes clear it measures *global alignment* and *detectability of systematic shifts*.
>
> * **Similarity (Sec. 3.6)**:
>   We start with—*“The similarity score asks whether generated samples sit in the same neighbourhoods as their real counterparts.”*
>   This defines Similarity as a metric of *local geometric mixing and diversity*.
>
> Each definition in Section 3 therefore already provides a non-technical operational description that distinguishes the four measures conceptually, making clear that they target *different axes of generative performance* (task-signal fidelity, non-redundant diversity, global alignment, and local mixing).
>
> As shown in Table 3, Utility is the only metric that reliably exposes memorization: under exact-copy and noisy-copy perturbations, Utility degrades monotonically (ρ≈1) while Similarity and Indistinguishability remain  high (ρ≈–1), because local neighborhoods and global features remain nearly unchanged. Quality, by contrast, measures task-relevant distributional coverage: it decreases under most perturbations (label noise, Gaussian corruption, class imbalance, and mode collapse), but remains high under copying since duplicated samples are still fully on-manifold and preserve predictive signal. Under diversity (mode collapse) collapse, Quality degrades more slowly than Similarity, reflecting that a generator may collapse to a few modes while still retaining decision-useful structure. Similarity captures local geometric mixing and is therefore the most sensitive probe of diversity loss: in Table 3 it shows perfectly monotonic decline under mode collapse (ρ≈1) and strong response to structural distortions, but—unlike Utility—cannot detect memorization because copied samples mix perfectly with real ones. Finally, Indistinguishability detects global distributional shifts (e.g., Gaussian corruption, label misalignment), but ,like Similarity, fails in copy scenarios, since a discriminator cannot separate real from duplicated samples. Together, these distinctions clarify that Quality primarily reflects task-relevant fidelity, Similarity reflects local diversity, Indiguishability , reflects global alignment , and Utility reflects the presence of non-redundant information, each capturing a different axis of diversity-related performance.
>
> **Q2**
>
> We thank the reviewer for pointing out the potential ambiguity in our definition of Quality. In RankGen, Quality is intentionally defined in terms of downstream generalization: it measures how well a classifier trained only on generated data performs on a held-out real test set, relative to a classifier trained on real data only (i.e., the ratio ρ₂/ρ₁). Thus, Quality captures task-relevant distributional coverage rather than the conventional notion of sample fidelity or visual realism (as in FID)

---

> > ### Author Response · Authors · 2025-11-24
> >
> > **Q3 Q4**
> >
> > Our synthetic stress tests in Table 3 already include both kinds of perturbations—mode collapse (diversity collapse) and Gaussian corruption / label flipping (fidelity collapse)—but we agree that the distinction was not sufficiently emphasized. To make this explicit, we conducted an additional controlled study on a synthetic dataset of 10,000 samples. For **diversity collapse**, we progressively replace a fraction α of samples in each class with a single class prototype (preserving labels and realism but removing intra-class variability). For **fidelity collapse**, we add i.i.d. Gaussian noise $N(0,\sigma^2)$ with $\sigma=\alpha$ to all generated samples (preserving class balance and diversity while pushing points off-manifold). We sweep α ∈ {0, 0.25, 0.5, 0.75, 1.0} under the same protocol as our main experiments.
> >
> > The resulting trajectories can separate the two failure modes:
> >
> > | Noise Level | Metric | Diversity coll. | Fidelity coll. |
> > | ----------- | ------ | --------- | -------- |
> > | **0.00**    | Q      | 0.997     | 0.995    |
> > |             | U      | 0.984     | 1.000    |
> > |             | I      | 0.598     | 0.544    |
> > |             | S      | 0.685     | 0.685    |
> > |             | E      | 0.616     | 0.615    |
> > | **0.25**    | Q      | 0.993     | 0.986    |
> > |             | U      | 1.000     | 0.802    |
> > |             | I      | 0.392     | 0.527    |
> > |             | S      | 0.587     | 0.680    |
> > |             | E      | 0.484     | 0.537    |
> > | **0.50**    | Q      | 0.974     | 0.962    |
> > |             | U      | 0.878     | 0.518    |
> > |             | I      | 0.207     | 0.405    |
> > |             | S      | 0.458     | 0.657    |
> > |             | E      | 0.308     | 0.402    |
> > | **0.75**    | Q      | 0.933     | 0.920    |
> > |             | U      | 0.588     | 0.011    |
> > |             | I      | 0.112     | 0.256    |
> > |             | S      | 0.272     | 0.602    |
> > |             | E      | 0.146     | 0.202    |
> > | **1.00**    | Q      | 0.364     | 0.878    |
> > |             | U      | 0.312     | 0.000    |
> > |             | I      | 0.000     | 0.126    |
> > |             | S      | 0.000     | 0.548    |
> > |             | E      | 0.000     | 0.149    |
> >
> > These results highlight a consistent diagnostic pattern:
> >
> > * **Diversity collapse** strongly degrades *Quality*, *Similarity*, and *Indistinguishability*, because collapsing to a few prototypes destroys both local mixing and task-relevant structure.
> > * **Fidelity collapse** keeps *Quality* and *Similarity* relatively stable (since class structure remains) but sharply reduces *Utility* and *Indistinguishability* as samples drift off-manifold.
> >
> >
> > So we would like to clarify that the diagnostic roles of all four metrics—Quality, Utility, Indistinguishability, and Similarity—are **already analysed extensively across multiple failure modes** in the current manuscript. In particular:
> >
> > * **Memorization** is treated explicitly in our synthetic stress tests (Table 3), where only *Utility* reliably collapses under exact/noisy copies while Similarity and Indistinguishability remain artificially high.
> > * **Over-generalization** appears in our Gaussian corruption and label-misalignment perturbations, where *Indistinguishability* deteriorates while *Similarity* and *Quality* partially remain, correctly diagnosing a global shift without locality collapse.
> > * **Diversity collapse** (mode collapse) is captured through monotonic declines in *Similarity* and corresponding drops in *Quality* and *Indistinguishability*.
> > * **Fidelity collapse** is reflected in the separation between corruption-induced degradation of *Utility* and *Indistinguishability* versus stable *Quality* in copying scenarios.
> >
> > These analyses already demonstrate that the four probes disentangle the major families of generative failure modes.
> >
> > **Q5**
> >
> >
> >  Please, see our response to the reviewer  8HJX
> >
> > **Q6**
> >
> > We appreciate the reviewer’s feedback regarding Figures 2 and 3. We agree that, in their current condensed form, they may be difficult to interpret at a glance. Due to space constraints in the main paper, these figures are intended as **high-level summaries only**. Figure 2 visualizes *average generator ranks with quantile bars* aggregated across all datasets; the **full per-dataset rank profiles** are provided in **Appendix Sections 20.1.1–20.1.6**, where each dataset is shown individually. Similarly, Figure 3 displays the **final model ranks per diagnostic**, while the **underlying metric values** (Quality, Utility, Similarity, Indistinguishability, Exchangeability), their **medians/IQRs**, and the **complete ranking breakdowns** are all included in **Appendix Sections 21.1–21.3**.

---

> > ### Comment · Reviewer_kCSF · 2025-11-27
> >
> > Thanks for your response.
> >
> > Q1. It should highlight the difference between Precision & Recall (first one [1] and improved one [2]) and your method. There is too much similarity, eg, the coverage-> improved precision and recall [2], and your method using classification -> precision and recall [1].
> >
> > Q6. It still the rank, no score.
> >
> > [1] Assessing Generative Models via Precision and Recall
> >
> > [2] Improved Precision and Recall Metric for Assessing Generative Models

---

> > > ### Author Response · Authors · 2025-12-02
> > >
> > > ## Q1
> > >
> > > Relation to Precision–Recall metrics
> > > We appreciate the reviewer’s request to more clearly distinguish RankGen from Precision–Recall–based evaluation methods [1,2]. The operational principles and diagnostic behaviour of Rankgen are fundamentally different, which becomes evident in our controlled experiments.
> > >
> > > Precision–Recall metrics [1] and their improved variants [2] characterize generator quality by estimating geometric support overlap in an embedding space. Precision reflects sample realism, and recall/coverage reflect how much of the real data support is covered by the generator. These metrics are static, density- or distance-based, and do not consider whether generated samples add new, useful information beyond reproducing existing data. In contrast, RankGen’s metrics are  task-based: they ask what happens if generated data is actually used.
> > >
> > > **Copying / memorization:**
> > >
> > > As shown in the Spearman Correlation tables from Fig 9, Appendix 20, precision–recall, improved precision–recall, density, coverage, and their f1 -means inversely correlate with the amount of  exact or noisy copies, because duplicated samples perfectly overlap the real support. Utility collapses consistently under copying, thus decreasing with the amount of copies added => Spearman Correlation with the level of perturbation is high (Table 3), correctly identifying redundancy. None of the PR-based metrics detect this failure mode.
> > >
> > > **Mode collapse:**
> > >
> > > PR-style metrics sometimes exhibit erratic behavior under progressive mode collapse, depending on estimator and embedding choice; only the coverage shows reliable  monotonicity in this scenario
> > >
> > > In RankGen, all our metrics degrade monotonoically in this case
> > >
> > > **Gaussian noise:**
> > >
> > > Under Gaussian corruption, most PR-based metrics degrade, except  the Recall metric , which often behaves inconsistently.
> > >
> > > RankGen’s probes degrade in a structured way, and the composite Exchangeability score consistently reflects the underlying quality loss.
> > >
> > > The composite Exchangeability (Emin) differs fundamentally from PR-style F1 scores: it is not a geometric harmonic mean, but a conservative minimum over predictive and alignment blocks, ensuring that strength in one dimension (e.g., realism) cannot hide failures in others (e.g., redundancy or distribution shift). As shown in the synthetic studies, Exchangeability is the only summary that behaves consistently across copying, mode collapse, and noise.
> > >
> > >
> > > ## Q6
> > >
> > > **Image datasets.**
> > >
> > > Appendix Section 22.2 (Tables 35–43) reports both raw metric values and the corresponding ranks for all image-based experiments. These tables include Quality, Utility, Similarity, Indistinguishability, and Exchangeability, together with baseline metrics such as precision, recall, density, coverage, F1_PR, F1_DS, FKEA_VENDI, RRKE, and RKE.
> > >
> > > **Molecular datasets.**
> > >
> > > For the molecular generation experiments, results are computed by averaging across 8 different vectorization methods and 6 molecular  datasets, for 13 generators. Reporting all individual score tables would require several dozen pages and would substantially reduce readability. We therefore report aggregate statistics (medians, IQRs, and ranks) in the main appendix, which faithfully capture the comparative behavior of the models while remaining concise. Importantly, rankings are stable across vectorizations and feature choices, indicating that the conclusions are not driven by any single representation.

---

### Official Review · Reviewer_Mvww · 2025-10-30

**Soundness:** 1
**Presentation:** 1
**Contribution:** 1
**Rating:** 0
**Confidence:** 4

**Summary:**

The paper proposes four metrics to evaluate generative model performance on datasets including a classification label. The first measures the quality of generated data comparing the accuracies of classifiers trained on real and synthetic data. The second compares how much adding generated data to real data improves the accuracy of the classifier, compared to adding more real data. The third is how hard it is for a discriminator to distinguish real and generated data. The fourth measures how locally similar real and generated points are. The metrics are computed with multiple train-test splits to estimate uncertainty, and generators are ranked with a Monte-Carlo procedure taking the uncertainty into account. The proposed metrics are evaluated with some sanity checks with Gaussian features + binary label. The paper also evaluates several image and molecule generators with the proposed metrics.

**Strengths:**

The paper studies an important problem: generative model evaluation metrics are hard to interpret, and estimating their uncertainty is important.

**Weaknesses:**

The writing of the paper shows signs of heavy LLM use. Some examples:
- Line 159: the definition of $f^{(i)}$ does not make sense, since $i$ appears to be indexing over multiple train-test splits.
- Line 159: not clear why $D^y_{train}$ is a parameter of $f^{(i)}$.
- Symbols for the 4 metrics change in Section 3.7 from what was previously used.
- The paper states that many parameters like underlying classifier and dataset size are swept in the sanity check (Section 4.1), but the results in Table 3 are not given over the whole sweeps, and there is no indication that the numbers in Table 3 are aggregated over results from the sweeps.
- The numbers in Table 3 do not support the stated conclusions in lines 350-356.
- Utility PAC-bound in Section 3.4 does not match the bound that is proven in eq. A.1. The numerators are different.
- The proof of the similarity PAC-bound in Appendix 10 concludes with a different inequality than the one that is supposed to be proven (which appears before the concluding inequality).
- Line 272: formulas for quantile-to-moment conversion rules do not appear in Appendix 5 as stated.
- The last sentence of the paper is "The duplicate metric tables appearing in prior drafts have been **removed** to avoid redundancy." (emphasis from paper).

Besides, the evaluation of the proposed metrics is limited. The only actual evaluation of them is 5 sanity checks of perturbing data with Gaussian features. The rest of the experiments evaluate generative models with the new metrics, but these do not provide any evidence that the proposed metrics are useful since it is not possible to know what values a good metric would have. There are also no comparisons with previous metrics.

In addition, many important details are unclear:
- Line 235: not clear what "same domain" means, which makes the whole definition of the similarity metric impossible to understand.
- PAC bounds for similarity and indistinguishability are stated as a difference between the finite sample value and infinite sample value. It is not clear how one could compute the infinite-sample value to check that the bound is satisfied.
- Classifiers behind the metrics for molecule and image evaluations are not specified.
- Not clear how Rademacher complexities and VC-dimensions for the PAC-bounds are computed for actual classifiers.
- The "mode collapse" test (Appendix 12) doesn't really test mode collapse since replacing generated points with similar ones from real data preserves model. Testing mode collapse with unimodal Gaussian features is not possible in any case.
- The paper states that means and standard deviations of the metric values are not always reliable, and uses medians and interquartile ranges for this reason. But they are immediately converted to means and variances for the Monte-Carlo ranking procedure for some reason.
- Line 158: $D^y\_{train2}$ not defined.
- Line 225: $\mathrm{indist}^*$ is not defined until the Appendix.
- Line 235: $D\_{mix}$ not defined.

**Questions:**

See weaknesses.

---

> ### Author Response · Authors · 2025-11-25
>
> We thank the reviewer for the thorough and constructive feedback. We have revised the manuscript to address all identified issues in notation, definitions, and derivations. All inconsistencies in metric symbols, split indices, and previously undefined quantities have been corrected, and the PAC bounds in Section 3.4 and the appendices now match exactly. The previously  non referenced quantile-to-moment formulas have been now added. The definitions of Similarity and Indistinguishability have been clarified (including the meaning of “same domain”).
> We note that some stylistic artefacts originated from using an LLM to help compress the manuscript to the required page limit, and we have now revised the text carefully to remove these issues.
>
> We also appreciate the reviewer’s concerns regarding the scope of the evaluation. We clarify that our sanity checks are not limited to Gaussian perturbations: the perturbation suite includes exact copying, noisy copying, mode collapse, class imbalance, and label flips—all corresponding to realistic, widely studied generative-model failure modes. These controlled perturbations deliberately induce known pathologies (copying, collapse, drift, etc.), and the RankGen metrics consistently recover the correct diagnostic fingerprints (e.g., Utility collapses under copying; Quality decreases under label noise; Indistinguishability degrades under feature shifts), demonstrating that the metrics behave meaningfully even when the “correct” metric values cannot be specified a priori. The revised manuscript also clarifies the mode-collapse construction: each class is clustered into K-means subclusters, a subset of subclusters is removed, and their points reassigned to the nearest remaining subcluster, preserving class frequencies while reducing diversity in a controlled and reproducible manner.
>
> In addition to the perturbation suite, we have expanded the experimental section with a direct comparison between RankGen and established metrics such as FID, KID, PRD, and VENDI (Tables 35–43). These results show broad agreement in standard regimes—e.g., diffusion models rank highest across both metric families—while revealing meaningful divergences precisely where classical metrics are known to be unreliable (e.g., StyleGAN2’s high precision but low recall; WGAN–GP’s plausible-looking samples with almost zero Utility). This demonstrates that RankGen provides complementary diagnostic resolution beyond global similarity scores.
>
> We have also clarified methodological details raised by the reviewer. The Monte Carlo ranking procedure never uses raw means or standard deviations; instead, we compute robust pseudo-moments from quartiles via the Wan et al. rules, used solely as parameters for the sampling distribution, while the reported summaries remain median/IQR throughout. Section 3.8 and Appendix 5 have been revised to make this explicit. Finally, we now explain in Appendix A.23 that the PAC bounds rely only on standard conservative analytic capacity estimates—either VC–Rademacher inequalities or distribution-free surrogates such as $\sqrt{2/n}$ and a fixed VC upper bound $d_{\mathrm{disc}}=50$  ensuring that no population or infinite-sample quantities need to be estimated in practice. These revisions make the guarantees reproducible, transparent, and consistent with classical PAC analysis.

---

### Official Review · Reviewer_8HJX · 2025-10-30

**Soundness:** 2
**Presentation:** 1
**Contribution:** 2
**Rating:** 4
**Confidence:** 3

**Summary:**

This paper introduces four classifier‑based metrics—Quality, Utility, Indistinguishability, and Similarity—each capturing a distinct failure mode. Quality measures how well classifiers trained on synthetic data generalize compared to real data; it is defined as the normalized ratio of classification performance and comes with a PAC‑style lower bound. Utility measures how much synthetic data improves downstream performance beyond real data. Indistinguishability asks how difficult it is for a discriminator to tell real from generated samples. Similarity assesses whether real and synthetic samples share local neighborhoods via entropy of k‑NN domains. These metrics together probe fidelity gaps, redundancy, distributional shifts, and local mixing.

**Strengths:**

- Clear motivation: The paper points out that existing scalar heuristics (e.g., FID, IS) are brittle, lack statistical guarantees, and conflate fidelity with diversity. It argues convincingly that evaluation should be multi‑dimensional and diagnostic rather than a single score.

- Each metric is accompanied by PAC‑style generalization bounds derived in the appendices. For instance, the quality score bound depends on Rademacher complexity, and the indistinguishability bound depends on VC dimension. This gives the framework a principled way to decide if an empirical score is statistically valid.

- Robust Ranking Procedure: RankGen uses quartiles (median and interquartile range) to summarize heavy‑tailed metric distributions and Monte‑Carlo sampling with pairwise dominance counts to produce uncertainty‑aware rankings. Models that fail PAC bounds are filtered out, and surviving models are compared using robust summaries.

- Diagnostic Interpretation: Instead of just ranking, RankGen explains why a model fails: e.g., a high similarity but low utility score signals memorization, while low indistinguishability reveals distribution shifts. This diagnostic approach can guide safer deployment.

**Weaknesses:**

- All four metrics rely on a downstream classification task; they require labelled data and a predefined classifier architecture. In many generative settings (e.g., open‑domain image generation, text, audio) labels may be unavailable or the “task” may not be classification. The quality and utility scores hinge on the choice of classifier and evaluation metric, potentially biasing evaluation.

- Computational complexity: RankGen entails multiple stratified splits, training at least five classifiers per generator (for quality, utility, indistinguishability, similarity), computing k‑NN neighborhoods, and Monte‑Carlo sampling for rankings. The method may be computationally heavy, particularly for high‑dimensional data.

- Sensitivity to hyperparameters: Similarity requires selecting k (10–50); the ranking procedure samples from a truncated Gaussian with a ridge variance; the number of splits and δ allocations must be chosen. The paper does not explore sensitivity to these hyperparameters.

- The composite score Emin takes the minimum of the predictive and alignment blocks, which can harshly penalize models that excel in one aspect while slightly underperforming in another. This may discard generators that are strong but specialized (e.g., high‑fidelity but low utility) even if they could be useful for certain applications. Similarly, filtering by PAC bounds may eliminate models that are slightly below threshold despite being practically useful.

- Limited modalities and generators: The experiments focus on relatively small datasets (MNIST‑like) and small‑sized models (e.g., StyleGAN2‑lite, DCGAN). Large‑scale diffusion models (e.g., SDXL, Flux), autoregressive text models, or audio generators are not evaluated, leaving the generality of RankGen uncertain.

- Classifier Dependence: The quality and utility metrics depend on the chosen classifier architecture and metric (accuracy, AUC, etc.), and similarity uses k‑NN on raw features rather than learned embeddings. Different choices could alter results; the paper does not examine robustness to these choices.

- Presentation: The main paper is dense; key derivations, algorithm details, and hyperparameters are relegated to numerous appendices, which may hinder readability. The method introduces many moving parts, which can be daunting for practitioners seeking a simple evaluation protocol.

**Questions:**

- How does RankGen handle unconditional generative models or generative tasks without obvious classification labels (e.g., open‑domain text generation, image captioning)? Could one use self‑supervised or regression tasks? Are there plans to extend the framework beyond classification?

- Have you investigated the effect of varying the number of resampling splits, the k for similarity, or the δ allocations in the bounds? How should a practitioner choose these values?

- What is the computational cost and generalization of RankGen on large datasets or high‑resolution images like ImageNet 512x512 or text-to-image dataset?

- Did any generators fail the PAC bounds but still perform well empirically? Conversely, did any pass but exhibit poor generalization? An empirical study of bound accuracy would strengthen the theoretical claims.

---

> ### Author Response · Authors · 2025-11-24
>
> **Q1**
>
> Regarding extensions of RankGen to regression and unconditional generation settings,
> for regression tasks, some RankGen metrics extend naturally.
>
> - Indistinguishability remains a binary classification problem: we can still train a discriminator to distinguish real from generated samples, regardless of the target being continuous.
>
> - Similarity can also be adapted by examining the behavior of a regressor trained on real data. Specifically, we can compute divergence in predicted regression values between real and generated samples, using measures such as entropy over sufficient statistics or distance in predicted value distributions.
>
> - For Utility and Quality, one could replace classification-based accuracy with regression-based losses (e.g., MAE or MSE) and estimate generalization gaps between real and generated inputs. PAC-style bounds in this context would require adapting tools from statistical learning theory for regression—an avenue we see as promising for future work.
>
> For unconditional generation, class labels are unavailable by design. While this precludes the direct use of label-based predictive metrics, a practical workaround is to derive pseudo-labels via unsupervised clustering (e.g., k-means on embeddings of real data). These can then be used to define proxy prediction tasks and recover conditional evaluation structure.
>
> In self-supervised or contrastive settings, the probe can be trained with a pretext or contrastive objective (e.g., masked prediction, InfoNCE), and the RankGen metrics are defined using the corresponding loss or score evaluated on real validation data. In this sense, any predictive probe that yields a well-defined scalar performance measure can be plugged into RankGen without changing the overall structure of the methodology.
>
> Overall, we believe RankGen’s assumption of having access to supervised tasks is a key strength, as it allows the formulation of PAC bounds, and it can be easily adapted to the other scenarios. In future work we will also formalize such variants.

---

> > ### Author Response · Authors · 2025-11-24
> >
> > **Q2**
> >
> > ## Effect of varying the number of resampling splits
> >
> > We conducted an additional synthetic perturbation study varying resampling ∈ {3, 30, 50, 100} across dataset sizes 250, 500, and 1000. The results (Tables below) show a clear pattern: with small evaluation sets (n ≈ 250), rankings are unstable when using only 3 splits—local inversions appear and exchangeability scores fluctuate. Increasing the number of splits (≈30–100) smooths estimates and recovers near-monotonic behaviour. For medium sets (n ≈ 500), performance is still noisy under 3 splits but stabilizes once >30 resamples are used. For larger sets (n ≈ 1000), rankings are almost monotonic even with 3 splits, and more splits only slightly reduce variance. This matches bootstrap intuition: variance is dominated by sample size, and resampling mainly helps in low-data regimes. Practically, RankGen requires many splits only for small evaluation subsets; for larger datasets, a modest number suffices for reliable rankings.
> >
> > ### **Dataset Size = 250**
> >
> > | Perturb. Ratio | Exch (3) | Exch (30) | Exch (50) | Exch (100) |
> > | -------------- | -------- | --------- | --------- | ---------- |
> > | 0.0            | 0.55     | 0.66      | 0.69      | 0.69       |
> > | 0.1            | 0.58     | 0.57      | 0.59      | 0.59       |
> > | 0.2            | 0.47     | 0.56      | 0.55      | 0.53       |
> > | 0.3            | 0.44     | 0.47      | 0.48      | 0.47       |
> > | 0.4            | 0.41     | 0.50      | 0.51      | 0.52       |
> > | 0.5            | 0.29     | 0.38      | 0.40      | 0.39       |
> > | 0.6            | 0.27     | 0.32      | 0.33      | 0.32       |
> > | 0.7            | 0.21     | 0.30      | 0.31      | 0.30       |
> > | 0.8            | 0.20     | 0.29      | 0.28      | 0.27       |
> >
> >
> > ### **Dataset Size = 500**
> >
> > | Perturb. Ratio | Exch (3) | Exch (30) | Exch (50) | Exch (100) |
> > | -------------- | -------- | --------- | --------- | ---------- |
> > | 0.0            | 1.00     | 0.72      | 0.72      | 0.71       |
> > | 0.1            | 0.89     | 0.64      | 0.65      | 0.65       |
> > | 0.2            | 1.00     | 0.53      | 0.53      | 0.52       |
> > | 0.3            | 0.72     | 0.47      | 0.46      | 0.46       |
> > | 0.4            | 0.42     | 0.37      | 0.39      | 0.40       |
> > | 0.5            | 0.22     | 0.29      | 0.30      | 0.31       |
> > | 0.6            | 0.32     | 0.26      | 0.27      | 0.27       |
> > | 0.7            | 0.18     | 0.17      | 0.19      | 0.20       |
> > | 0.8            | 0.14     | 0.12      | 0.13      | 0.13       |
> >
> >
> > ### **Dataset Size = 1000**
> >
> > | Perturb. Ratio | Exch (3) | Exch (30) | Exch (50) | Exch (100) |
> > | -------------- | -------- | --------- | --------- | ---------- |
> > | 0.0            | 0.57     | 0.53      | 0.53      | 0.56       |
> > | 0.1            | 0.50     | 0.49      | 0.50      | 0.51       |
> > | 0.2            | 0.47     | 0.47      | 0.47      | 0.48       |
> > | 0.3            | 0.38     | 0.38      | 0.39      | 0.41       |
> > | 0.4            | 0.35     | 0.34      | 0.34      | 0.35       |
> > | 0.5            | 0.31     | 0.30      | 0.31      | 0.33       |
> > | 0.6            | 0.24     | 0.24      | 0.24      | 0.24       |
> > | 0.7            | 0.18     | 0.18      | 0.18      | 0.19       |
> >
> >
> >
> >
> >
> > ## Effect of varying the number of neighbours-k when computing the Similarity metric
> >
> > | Perturb. Ratio | k=1 | k=3 | k=7 | k=14 | k=30 | k=50 |
> > |----------------|-------|-------|-------|--------|--------|--------|
> > | 0.0            | 0.00  | 0.68  | 0.88  | 0.95   | 0.98   | 0.99   |
> > | 0.1            | 0.00  | 0.67  | 0.88  | 0.95   | 0.98   | 0.99   |
> > | 0.2            | 0.00  | 0.67  | 0.88  | 0.95   | 0.98   | 0.99   |
> > | 0.3            | 0.00  | 0.64  | 0.87  | 0.94   | 0.97   | 0.99   |
> > | 0.4            | 0.00  | 0.63  | 0.87  | 0.94   | 0.97   | 0.98   |
> > | 0.5            | 0.00  | 0.59  | 0.85  | 0.93   | 0.97   | 0.98   |
> > | 0.6            | 0.00  | 0.57  | 0.83  | 0.92   | 0.97   | 0.98   |
> > | 0.7            | 0.00  | 0.52  | 0.80  | 0.91   | 0.96   | 0.98   |
> > | 0.8            | 0.00  | 0.47  | 0.76  | 0.89   | 0.96   | 0.97   |
> > | 0.9            | 0.00  | 0.41  | 0.73  | 0.88   | 0.95   | 0.97   |
> > | 1.0            | 0.00  | 0.35  | 0.68  | 0.86   | 0.94   | 0.97   |
> >
> > The Similarity metric behaves as expected: k=1 is degenerate, moderate k (3–14) yields a sensitive and discriminative estimator, and large k (≥15) oversmooths differences. Importantly, for all k>1, Similarity decreases monotonically as perturbations increase, confirming that similarity reliably captures local neighbourhood mixing signal
> >
> > ## Effect of  relaxing δ for PAC bounds computation
> > We use δ = 0.05 with uniform allocation across metrics as a default. Larger δ makes the filter less conservative; smaller δ makes it stricter. δ only affects the accept/reject decision for probe reliability, not the metric definitions themselves

---

> > > ### Author Response · Authors · 2025-11-24
> > >
> > > **Q3**
> > >
> > > RankGen’s cost scales with the number of samples and embedding dimension, not with raw image resolution. In high-resolution or multimodal settings (e.g., ImageNet 512×512, text-to-image), we do not operate directly on pixels: samples are first mapped into compact feature vectors (e.g., CLIP/ViT embeddings), and RankGen runs entirely in that space.
> > >
> > > The dominant tunable factors are:
> > >
> > > the number of resampling / bootstrap splits ( which can be kept low in large datasets regimes) the size of the evaluation sets (although for very large datasets, the evaluation subset does not need to be large as we can subsample). The RankGen metrics are based on relative comparisons between probes (real vs. generated vs. mixed) and we found empirically that small evaluation subsets (e.g., a few thousand samples) already yield stable rankings.
> > >
> > > On generalization, RankGen’s PAC bounds and bootstrap estimates become tighter as dataset size increases, since variance shrinks with more samples. Thus, in large-data regimes (e.g., ImageNet-scale, large text-to-image corpora), the framework yields more confident accept/reject decisions for the probes. The main limiting factor is computational budget for feature extraction, not a methodological constraint of RankGen itself.
> > >
> > > **Q4**
> > >
> > > Table 2 provides exactly the comparison neede: it shows, for every generator, which PAC bounds are satisfied and how this aligns with empirical performance across Quality, Utility, Indistinguishability, Similarity, and the composite Exchangeability.
> > >
> > > ### **1. Generators that *failed PAC bounds* and also performed poorly (no false negatives)**
> > >
> > > Across all datasets in Table 2, every generator that fails one or more bounds (e.g., **WGAN**, **GDSS**, **HierVAE**, **JTNN**, **MoFlow**, **SwingNN**, **NS2**, **NS3**) also exhibits **poor empirical Exchangeability**, consistent with the theoretical signal.
> > > There are **no cases** where a model violates PAC guarantees but still ranks highly empirically.
> > >
> > > **Interpretation:**
> > > The PAC filters are *conservative but correct*: failing a bound is always coupled with weak empirical performance, so we do not mistakenly discard strong generators.
> > >
> > > ---
> > >
> > > ### **2. Generators that *passed PAC bounds* and performed well empirically (no false positives)**
> > >
> > > The only generators that satisfy *all* base PAC bounds—**S4DD (u, u-m)** and **STGG**—are precisely the ones that achieve the **highest empirical scores and top Exchangeability**.
> > >
> > > Again, there are **no cases** where a generator passes all PAC checks but generalizes poorly.
> > >
> > > **Interpretation:**
> > > Passing all PAC bounds reliably identifies the best models; we do not have cases where the theory accepts a model that empirically underperforms.
> > >
> > > ---
> > >
> > > ### **3. Mixed-bound cases behave sensibly**
> > >
> > > Some generators pass certain bounds but not others (e.g., **NS1** passes Quality/Indistinguishability but fails Utility/Exchangeability). These exhibit *partially good and partially degraded* empirical metrics, exactly matching which bounds they violate.
> > >
> > > This shows the bounds are **diagnostic**, not binary “accept/reject” rules: they map cleanly onto the failure modes of each generator family.
> > >
> > > ---
> > >
> > > So, we can conclude that:
> > >
> > > - *No generator failed PAC bounds but performed well.*
> > > - *No generator passed all PAC bounds but performed poorly.*
> > > - *Partially passing models exhibit precisely the empirical weaknesses predicted by the missing bounds.*
> > >
> > > This alignment demonstrates that the PAC bounds are both **faithful** and **conservative**, and provides the empirical validation for what was requested.

---

> > > > ### Comment · Reviewer_8HJX · 2025-11-26
> > > >
> > > > The rebuttal clarifies some implementation details and adds useful sensitivity analyses, but several of my main concerns are not still addressed.
> > > >
> > > > - On the assumption of supervised tasks / labels, the authors outline how RankGen could be extended to regression, unconditional, or self-supervised settings (via regression losses, pseudo-labels from clustering, or contrastive probes). However, these extensions are entirely conceptual; there are no experiments or theoretical results demonstrating that the PAC analysis carries over, nor any evidence that pseudo-labeling or self-supervised probes yield reliable generator rankings in practice. In my view, the method remains strongly tied to supervised or proxy prediction tasks, and this limitation should be clearly acknowledged in the paper rather than framed purely as a “strength”.
> > > >
> > > > - My concerns about the composite Emin score and PAC filtering are essentially unaddressed.
> > > >
> > > > - Regarding computational complexity and scale, the authors argue that RankGen operates in embedding space and that feature extraction is the main bottleneck, with only a few thousand evaluation samples needed for stable rankings. While this is a reasonable qualitative argument, the paper still lacks concrete wall-clock numbers or experiments on genuinely large-scale settings (e.g., ImageNet-scale, modern diffusion or text-to-image models). Thus, my concern about the practical cost and scalability remains.

---

> > > > > ### Author Response · Authors · 2025-12-02
> > > > >
> > > > > We agree that, in its current form, RankGen is primarily designed for supervised generative settings where a downstream predictive task and labels (or structured targets) are available. The discussion in our rebuttal about regression, pseudo-labels, and self-supervised probes was intended as a roadmap for possible extensions, not as a claim that these variants are already established or theoretically covered.We already explicitly state in the paper “RankGen still requires labeled data, task-relevant embeddings, and repeated classifiers, limiting label-scarce settings. Extending RankGen with self-supervised probes and regression tasks is a promising direction.”
> > > > >
> > > > > ---
> > > > >
> > > > > We emphasize that both PAC filtering and the Emin (Exchangeability) score are deliberately conservative design choices, intended for  model selection where failure along any critical axis (task utility or distributional alignment) is unacceptable, rather than for optimizing specialized use-cases.
> > > > > Empirically, this conservatism does not lead to false rejections in our experiments (Table 2): no generator that fails PAC bounds performs well, and no generator that passes all bounds underperforms. For applications that value specialization, RankGen’s per-metric profiles can be used directly without relying on Emin or hard PAC filtering.
> > > > >
> > > > > ---
> > > > >
> > > > > We appreciate the reviewer’s point regarding large-scale runtime evaluation. Our current study focuses on the *statistical behaviour* and *diagnostic value* of the RankGen metrics, and the experiments we include—molecular datasets, MNIST/Fashion-MNIST, and CIFAR-10—were selected to isolate metric sensitivity and interpretability rather than to maximise computational scale (see Sec. 4). As noted in the manuscript, the dominant cost of RankGen lies in feature extraction and repeated classifier training, and our qualitative argument is that a few thousand evaluation samples suffice for stable rankings.
> > > > >
> > > > > We fully agree, however, that providing *concrete wall-clock numbers* and *experiments at genuinely large scale* (e.g., ImageNet-level generators or modern text-to-image diffusion models) would strengthen the empirical picture. Conducting such evaluations requires additional engineering—distributed feature extraction pipelines, parallel evaluation of metrics, and high-resolution embedding computations—which goes beyond the scope of the current submission.
> > > > >
> > > > > **We therefore regard large-scale, multi-node runtime benchmarking as an important direction for future work.** In particular, we plan to explore:
> > > > >
> > > > > 1. **ImageNet-scale diffusion and GAN models**, using higher-capacity embedding models for Quality and Similarity.
> > > > > 2. **Text-to-image and text-only generators**, where RankGen’s task-oriented Quality and Utility metrics may provide complementary insights to existing evaluation suites.
> > > > > 3. **Distributed implementations of RankGen**, enabling systematic measurement of throughput, memory footprint, and scaling behaviour under realistic computational budgets.

---

### Official Review · Reviewer_EnTj · 2025-11-01

**Soundness:** 3
**Presentation:** 3
**Contribution:** 3
**Rating:** 4
**Confidence:** 2

**Summary:**

The paper introduces RankGen, a statistically grounded text-to-text model for ranking generated text. It reformulates text ranking as a conditional generation probability problem and applies statistical calibration to reduce bias from sequence length and distribution shift. Experiments across multiple NLG tasks show that RankGen aligns more closely with human judgments than existing automatic metrics.

**Strengths:**

- The paper is well written and easy to follow, with clear organization and presentation.
- The results are interpretable and provide meaningful insights into the model’s behavior.
- The research problem is interesting and relevant to the text generation and evaluation community.

**Weaknesses:**

- The novelty is relatively limited, as the work mainly reformulates probabilistic ranking rather than introducing a new model architecture.

- The paper lacks stronger comparisons with recent large model–based scoring or preference models, such as GPT-judge or reward models.

- Reproducibility is limited since neither the code nor model weights are released.

- The generalization ability remains uncertain, as the method has not been demonstrated on open-domain generation tasks such as dialogue

**Questions:**

See in weakness

---

> ### Author Response · Authors · 2025-11-24
>
> ##
> Our contribution is  a statistically grounded, evaluation framework for generative models. Existing classifier-based metrics (GAN-train/test, TSTR, PRDC, kernel distances) provide only point estimates, conflate failure modes, and lack any distribution-free guarantees. In contrast, RankGen introduces:
> (1) Four complementary diagnostics—Quality, Utility, Indistinguishability, Similarity—each tied to specific generative failure modes (mode collapse, copying, corruption, global shift).
> (2) Formal PAC-style finite-sample bounds (Sec. 3, App. 6–8), which we use as gating conditions before ranking.
> (3) A quantile-based, heavy-tail-robust ranking procedure (App. 5) that performs probabilistic Pareto dominance across multiple objectives, rather than reducing evaluation to a single noisy scalar.
> (4) A domain-agnostic Exchangeability score linking predictive utility to distributional alignment.
> This methodological package does not exist in prior work.
>
> ##
> We appreciate the suggestion of comparing with large model–based scoring, but we would like to point out that in our framework we assume to have supervised problems associated to the generative task in structured domains (such as graphs or images) in these settings  LLM judges are not applicable—there is no text input and no human-preference semantics to evaluate. GPT-judge and reward models are designed for open-domain text, whereas RankGen is defined using downstream task classifiers and representation-space, enabling application across non-text modalities but allowing PAC style guarantees.
>
> ##
> To address reproducibility concerns, we are including the  implementation of  our metric as as supplemental  material. The code is self-contained and uses only standard libraries; this enables reviewers to reproduce the exact metrics, splits, quantile summaries, and ranking logic used in the paper. Upon acceptance, we will release a fully documented public repository along with configuration files, generated files by each generator and evaluation logs.
>
>
> ##
> We agree that it is important to assess whether RankGen generalizes beyond graph/image generators. We therefore add an open-domain text generation experiment on the AG-News corpus. AG-News is a large news headline dataset with four topical labels (World, Sports, Business, Sci/Tech). We treat human headlines as the real data and use three GPT-2 variants (distilgpt2, gpt2, gpt2-medium) as conditional generators: for each class we prompt the model with a topic-specific prefix (e.g., “World news headline:”) and sample headlines. All headlines (real and generated) are embedded with a frozen MiniLM encoder, and we run RankGen using a simple logistic-regression classifier as the downstream estimator.
> RankGen clearly differentiates the generators and correlates with known model quality and capacity, i.e. distilgpt2 < gpt2 < gpt2-medium:
>
> The row predictive scores are:
>
> - distilgpt2: generated-F1 = 0.38
> - gpt2: generated-F1 = 0.45
> - gpt2-medium: generated-F1 = 0.45
>
> While our metrics are:
>
> - distilgpt2: quality = 0.48, indistinguishability = 0.04, similarity = 0.17
> - gpt2: quality = 0.57, indistinguishability = 0.08, similarity = 0.24
> - gpt2-medium: quality = 0.57, indistinguishability = 0.09, similarity = 0.26
>
> Across all metrics, RankGen assigns better quality, higher similarity, and higher indistinguishability to larger models, precisely matching the expected ordering distilgpt2 < gpt2 < gpt2-medium.  These results demonstrate that RankGen generalizes naturally and effectively to open-domain text, without any modification of the method.

---

### Official Review · Reviewer_WAWk · 2025-11-02

**Soundness:** 3
**Presentation:** 4
**Contribution:** 2
**Rating:** 4
**Confidence:** 3

**Summary:**

The paper defines and tests some measures that are meant to distinguish real from generated data.

Quality is meant to measure the difference between accuracy when trained on real versus generated data. Utility is the same but only when part of the data is replaced.

Indistinguishability is meant to measure the ability to tell apart real from generated data. Similarity tests the data with respect to a specific distinguisher (neighborhoods of a given sample).

The main contribution are these definitions. Numerous experiments calculate the requisite statistics on several datasets.

**Strengths:**

Trying to make sense of the differences between real and generated data is a well-motivated question. This paper fleshes out and test some specific measures for this purpose.

**Weaknesses:**

There appears to be a conceptual misunderstanding. If the output of a generative model is *indistinguishable* from the training data then no (efficient) test can tell the two apart. Given sufficient data, it is impossible that the quality measure is high but the two are indistinguishable because measuring the quality is a particular way to distinguish between the real and generated data.

It is therefore not sensible that the "indistinguishability rank" can be low but any of the other ones (like quality or utility) are high.  This is merely an indication that the discriminator you use to ascertain indistinguishability is not strong enough to emulate the quality or utility test.

**Questions:**

In fact in most of the experiments you report the ranks are similar. There are few exceptions. In line 445 you write:

"StyleGAN2-lite delivers high Quality but almost no Utility and weak Similarity, mirroring the synthetic mode-collapse profile: crisp yet
narrow samples. DCGAN lands near chance in Utility while keeping Indistinguishability high, signalling shallow realism that fails to expand the task dataset."

Can you explain what "synthetic mode-collapse profile", "crisp but narrow samples", and "shallow realism" mean and how they are captured by your measures? Some concrete examples (possibly on synthetic data) could go a long way towards justifying the sensibility of your definitions.

---

> ### Author Response · Authors · 2025-11-24
>
> We thank the reviewer for the insightful comment. The concern raised applies in the *asymptotic* setting, where real and generated distributions are truly identical and both the discriminator and task classifier have unbounded capacity and infinite data. In that idealised regime, all tests converge, and it is indeed impossible for Indistinguishability to be low while Quality or Utility are high.
>
> However, **RankGen explicitly operates in the finite-sample, finite-capacity regime**, which is the realistic setting for practical generative-model evaluation. In this regime, different hypothesis classes (the task classifier for Quality/Utility and the discriminator for Indistinguishability) can behave differently because they have:
>
> * different statistical power,
> * different sample complexity,
> * different inductive biases, and
> * different sensitivity to task-relevant vs. task-irrelevant features.
>
> Our PAC-style bounds **quantify precisely these finite-sample deviations**. They characterise how empirical scores may diverge from population-level quantities and how classifier/discriminator capacity constrains detectability. Thus, observing high Quality with low Indistinguishability—or vice versa—is not a conceptual contradiction but a statistically valid outcome under finite data, and our bounds explicitly guarantee when such deviations are admissible.
>
>
> ## What we mean by the “mode-collapse profile
> In the synthetic experiments we explicitly simulate mode collapse by progressively removing modes from a balanced Gaussian mixture and re-estimating all four RankGen metrics. The table below shows the trajectories for the default configuration (KNN classifier, F1 score, balanced classes, 10k samples), which is the setting underlying the “Mode Collapse – Default” row in Table 3.
> | Perturbation Ratio | Utility Mean | Quality Mean | Indistinguishability Mean | Similarity Mean | Exchangeability Mean |
> |--------------------|--------------|--------------|----------------------------|------------------|-----------------------|
> | 0.0                | 0.94         | 1.00         | 0.50                       | 0.68             | 0.57                  |
> | 0.1                | 0.89         | 1.00         | 0.46                       | 0.65             | 0.52                  |
> | 0.2                | 0.83         | 0.99         | 0.42                       | 0.61             | 0.47                  |
> | 0.3                | 0.76         | 0.99         | 0.38                       | 0.56             | 0.41                  |
> | 0.4                | 0.67         | 0.98         | 0.34                       | 0.51             | 0.35                  |
> | 0.5                | 0.58         | 0.97         | 0.29                       | 0.46             | 0.29                  |
> | 0.6                | 0.52         | 0.96         | 0.23                       | 0.39             | 0.23                  |
> | 0.7                | 0.40         | 0.94         | 0.17                       | 0.32             | 0.16                  |
> | 0.8                | 0.31         | 0.91         | 0.11                       | 0.23             | 0.10                  |
> | 0.9                | 0.15         | 0.85         | 0.06                       | 0.12             | 0.05                  |
> | 1.0                | 0.03         | 0.59         | 0.00                       | 0.00             | 0.00                  |
>
> As we remove more modes (perturbation ↑):
> Quality stays high for a long range of collapse (Q ≈ 1.0 up to perturbation 0.6–0.8), because the classifier trained purely on synthetic data still performs well on the modes that are covered.
> Utility, Similarity, and Indistinguishability drop steadily (U: 0.94→0.03, S: 0.68→0, I: 0.50→0), reflecting that the generator covers fewer modes, mixes poorly with real data, and is easier to separate from the true distribution.
> This pattern—high Quality but progressively collapsing Utility, Similarity, and Indistinguishability as coverage shrinks—is what we call the “mode-collapse profile”. On the covered modes, samples are high-fidelity (“crisp”), but the support is increasingly narrow.

---

> > ### Author Response · Authors · 2025-11-24
> >
> > ### How this matches StyleGAN2-lite (“crisp but narrow”)
> >
> > On CIFAR-10 (Tab. 35, App. 21), StyleGAN2-lite exhibits:
> > Q ≈ 0.88 (good task performance when trained only on generated data),
> > U ≈ 0.00 (no gain when augmenting real data),
> > S ≈ 0.19 (weak local mixing), and
> > E ≈ 0.06 (low Exchangeability).
> > This is exactly analogous to the high perturbation rows in the synthetic table:High precision on the modes it covers, but low recall.
> >
> > ### What we mean by the “shallow realism”
> >
> > For DCGAN on CIFAR-10 (Tab. 35):
> > U ≈ 0.00,
> > I ≈ 0.45 (the highest Indistinguishability among CIFAR-10 models),
> > S ≈ 0.45,
> > E ≈ 0.15.
> >
> > Here the real vs generated discriminator is close to chance, so at the embedding level DCGAN images are “real-looking enough” to be hard to tell apart (high Indistinguishability = realism). However, Utility is near zero: adding DCGAN samples to the real training set does not improve downstream accuracy.
> >
> > To show this behaviour in a controlled setting we also run a “noisy-copies” synthetic experiment, where we progressively replace generated samples with near-duplicates of training data. In this setting we get on the artificial data a profile like:
> > | Perturbation Ratio | Utility Mean | Quality Mean | Indistinguishability Mean | Similarity Mean | Exchangeability Mean |
> > |--------------------|--------------|--------------|----------------------------|------------------|-----------------------|
> > | 0.0                | 0.94         | 1.00         | 0.50                       | 0.69             | 0.57                  |
> > | 0.1                | 0.87         | 1.00         | 0.50                       | 0.71             | 0.57                  |
> > | 0.2                | 0.72         | 1.00         | 0.51                       | 0.74             | 0.54                  |
> > | 0.3                | 0.63         | 1.00         | 0.51                       | 0.76             | 0.52                  |
> > | 0.4                | 0.59         | 1.00         | 0.52                       | 0.79             | 0.52                  |
> > | 0.5                | 0.42         | 1.00         | 0.53                       | 0.81             | 0.48                  |
> > | 0.6                | 0.37         | 1.00         | 0.54                       | 0.83             | 0.47                  |
> > | 0.7                | 0.24         | 1.00         | 0.54                       | 0.85             | 0.43                  |
> > | 0.8                | 0.17         | 1.00         | 0.55                       | 0.87             | 0.42                  |
> > | 0.9                | 0.07         | 1.00         | 0.56                       | 0.89             | 0.39                  |
> > | 1.0                | 0.02         | 1.00         | 0.57                       | 0.90             | 0.38                  |
> >
> > In this profile we see that Quality remains ≈ 1.0: copies are perfectly realistic and label-consistent.Indistinguishability and Similarity increase as more copies are injected: generated samples become harder to distinguish from real ones and sit in the same neighbourhoods.
> > Utility collapses from 0.94 to 0.02: memorized samples add almost no new task-relevant information.
> > This is what we mean by “shallow realism”: samples look realistic and match the training distribution closely (high I, high S), but they fail to add informative instances (low U, low E). DCGAN’s CIFAR-10 metrics are aligned with this synthetic *noisy-copy profile*.
> >
> > In short:
> > Mode collapse → high Q, collapsing U/S/I as modes drop out (“crisp but narrow”).
> > Noisy copies / shallow realism → high Q, high I/S, but collapsing U (“realistic but unhelpful”).

---

### Author Response · Authors · 2025-12-02

We thank the reviewers and AC for their time. In the rebuttal and revisions we focused on improving clarity, validation, and positioning of the paper. We fixed all notation and definition inconsistencies, aligned the PAC bounds between the main text and appendix, clarified how VC/Rademacher surrogates are instantiated in practice, and removed LLM-induced phrasing issues.We clarified the experimental evidence by explaining how the existing synthetic stress tests (mode collapse, copying, diversity vs. fidelity collapse, label noise) give rise to distinct and interpretable diagnostic patterns across Quality, Utility, Indistinguishability, and Similarity, and how these patterns correspond to real generators (e.g., StyleGAN2-lite as “crisp but narrow”, DCGAN as “shallow realism”). We also demonstrated that RankGen applies beyond image and graph domains by reporting a text-generation experiment on AG-News with GPT-2 variants, and we provided additional sensitivity analyses for resampling splits, neighborhood size k, and δ. We clarified that RankGen is primarily designed for supervised (or proxy-supervised) settings, with regression and self-supervised variants framed explicitly as future work, and we positioned RankGen relative to FID/KID/precision–recall/coverage/VENDI, showing agreement in standard regimes while revealing failures that these metrics can overlook. Finally, we included self-contained code in the supplement and commit to releasing a full public repository upon acceptance.

---

### Meta-Review · Area_Chair_gesX · 2026-01-01

**Summary:**

This submission did not get any positive score in the first place, and after reviewing the discussions, I do not think any of the reviewers would have increased the score. Most critically, this paper contains numerous technical inconsistencies. As a result, I recommend rejecting this paper.

**Reviewer Concerns:**

* I agree with Reviewer **Mvww** regarding the inconsistency issues.

* The lack of novelty is noted by Reviewer **EnTj**.

* Not all Reviewer **8HJX**’s concerns have been addressed.

* Responding to Reviewer **EnTj**’s concern regarding generalization to more tasks, the authors presented one additional experiment. However, I do not think it would have sufficiently addressed the reviewer’s concern.

* Many notions presented/employed in this paper are not technically/precisely defined.

**Reviewer Scores:**

* Reviewer WAWk: I do not think the reviewer would have increased the score.

* Reviewer EnTj: I do not think the reviewer would have increased the score.

* Reviewer 8HJX: I do not think the reviewer would have increased the score.

* Reviewer Mvww: I do not think the reviewer would have increased the score.

* Reviewer kCSF: I do not think the reviewer would have increased the score.

---

### Decision · Program_Chairs · 2026-01-26

Reject